# On the Expressiveness of Approximate Inference in Bayesian Neural Networks

**Andrew Y. K. Foong**[*]
University of Cambridge
ykf21@cam.ac.uk

**David R. Burt**[*]
University of Cambridge
drb62@cam.ac.uk

**Yingzhen Li**
Microsoft Research
Yingzhen.Li@microsoft.com

**Richard E. Turner**
University of Cambridge
Microsoft Research
ret26@cam.ac.uk

## Abstract

While Bayesian neural networks (BNNs) hold the promise of being flexible, well-calibrated statistical models, inference often requires approximations whose consequences are poorly understood. We study the quality of common variational methods in approximating the Bayesian predictive distribution. For single-hidden layer ReLU BNNs, we prove a fundamental limitation in *function-space* of two of the most commonly used distributions defined in *weight-space*: mean-field Gaussian and Monte Carlo dropout. We find there are simple cases where neither method can have substantially increased uncertainty in between well-separated regions of low uncertainty. We provide strong empirical evidence that exact inference does not have this pathology, hence it is due to the approximation and not the model. In contrast, for deep networks, we prove a universality result showing that there exist approximate posteriors in the above classes which provide flexible uncertainty estimates. However, we find empirically that pathologies of a similar form as in the single-hidden layer case can persist when performing variational inference in deeper networks. Our results motivate careful consideration of the implications of approximate inference methods in BNNs.

## 1 Introduction

Bayesian neural networks (BNNs) [27, 30] aim to combine the strong inductive biases and flexibility of neural networks (NNs) with the probabilistic framework for uncertainty quantification provided by Bayesian statistics. However, performing exact inference in BNNs is analytically intractable and requires approximations. A variety of scalable approximate inference techniques have been proposed, with mean-field variational inference (MFVI) [15, 4] and Monte Carlo dropout (MCDO) [11] among the most used methods. These methods have been succesful in applications such as active learning and out-of-distribution detection [9, 33]. However, it is unclear to what extent the successes (and failures) of BNNs are attributable to the exact Bayesian predictive, rather than the peculiarities of the approximation method. From a Bayesian modelling perspective, it is therefore crucial to ask, *does the approximate predictive distribution retain the qualitative features of the exact predictive?*

Frequently, BNN approximations define a simple class of distributions over the model parameters, (an *approximating family*), and choose a member of this family as an approximation to the posterior.

---

[*]Equal contribution.

Both MFVI and MCDO follow this paradigm. For such a method to succeed, two criteria must be met:

**Criterion 1** The approximating family must contain good approximations to the posterior.

**Criterion 2** The method must then select a good approximate posterior within this family.

For nearly all tasks, the performance of a BNN only depends on the distribution over weights to the extent that it affects the distribution over predictions (i.e. in 'function-space'). Hence for our purposes, a 'good' approximation is one that captures features of the exact posterior in function-space that are relevant to the task at hand. However, approximating families are often defined in weight-space for computational reasons. Evaluating **Criterion 1** therefore involves understanding how weight-space approximations translate to function-space, which is a non-trivial task for highly nonlinear models such as BNNs.

In this work we provide both theoretical and empirical analyses of the flexibility of the predictive mean and variance functions of approximate BNNs. Our major findings are:

1. For shallow BNNs, there exist simple situations where *no* mean-field Gaussian or MC dropout distribution can faithfully represent the exact posterior predictive uncertainty (**Criterion 1** is not satisfied). We prove in section 3 that in these instances the variance function of any fully-connected, single-hidden layer ReLU BNN using these families suffers a lack of '*in-between uncertainty*': increased uncertainty in between well-separated regions of low uncertainty. This is especially problematic for lower-dimensional data where we may expect some datapoints to be in between others. Examples include spatio-temporal data, or Bayesian optimisation for hyperparameter search, where we frequently wish to make predictions in unobserved regions in between observed regions. We verify that the exact posterior predictive does not have this limitation; hence this pathology is attributable solely to the restrictiveness of the approximating family.

2. In section 4 we prove a universal approximation result showing that the mean and variance functions of deep approximate BNNs using mean-field Gaussian or MCDO distributions can uniformly approximate any continuous function and any continuous non-negative function respectively. However, it remains to be shown that appropriate predictive means and variances will be found when optimising the ELBO. To test this, we focus on the low-dimensional, small data regime where comparisons to references for the exact posterior such as the limiting GP [30, 22, 28] are easier to make. In section 4.2 we provide empirical evidence that in spite of its theoretical flexibility, VI in deep BNNs can still lead to distributions that suffer from similar pathologies to the shallow case, i.e. **Criterion 2** is not satisfied.

In section 5, we provide an active learning case study on a real-world dataset showing how in-between uncertainty can be a crucial feature of the posterior predictive. In this case, we provide evidence that although the inductive biases of the BNN model with exact inference can bring considerable benefits, these are lost when MFVI or MCDO are used. Code to reproduce our experiments can be found at `https://github.com/cambridge-mlg/expressiveness-approx-bnns`.

## 2   Background

Consider a regression dataset $\mathcal{D} = \{(\mathbf{x}_n, y_n)\}_{n=1}^N$ with $\mathbf{x}_n \in \mathbb{R}^D$ and $y_n \in \mathbb{R}$. To define a BNN, we specify a prior distribution with density $p(\theta)$ over the NN parameters. Each parameter setting corresponds to a function $f_\theta : \mathbb{R}^D \to \mathbb{R}$. We specify a likelihood $p(\{y_n\}_{n=1}^N | \{\mathbf{x}_n\}_{n=1}^N, f_\theta)$ which describes the relationship between the observed data and the model parameters. The posterior distribution over parameters has density $p(\theta|\mathcal{D}) \propto p(\{y_n\}_{n=1}^N | \{\mathbf{x}_n\}_{n=1}^N, f_\theta) p(\theta)$. The posterior does not have a closed form and approximations must be made in order to make predictions.

### 2.1   Approximate Inference Methods

Many approximate inference algorithms define a parametric class of distributions, $\mathcal{Q}$, from which to select an approximation to the posterior. For BNNs, the distributions in $\mathcal{Q}$ are defined over the model parameters $\theta$. For example, $\mathcal{Q}$ may be the set of all fully-factorised Gaussian distributions, in which case the variational parameters $\phi$ are a vector of means and variances. We denote this family as $\mathcal{Q}_{\text{FFG}}$. A density $q_\phi(\theta) \in \mathcal{Q}$ is then chosen to best approximate the exact posterior according to some criteria. Once $q_\phi$ is selected, predictions at a test point $(\mathbf{x}_*, y_*)$ can be made by replacing the

expectation under the exact posterior by an expectation under the approximate posterior:

$$p(y_*|\mathbf{x}_*, \mathcal{D}) = \mathbb{E}_{p(\theta|\mathcal{D})}\left[p(y_*|\mathbf{x}_*, f_\theta)\right] \approx \mathbb{E}_{q_\phi(\theta)}\left[p(y_*|\mathbf{x}_*, f_\theta)\right] \approx \frac{1}{M}\sum_{m=1}^{M} p(y_*|\mathbf{x}_*, f_{\theta_m}), \quad (1)$$

where $\theta_m \sim q_\phi$ on the RHS of equation (1). Many approximate inference algorithms may share the same $\mathcal{Q}$, e.g. VI, the diagonal Laplace approximation [7], probabilistic backpropagation [13], stochastic expectation propagation [25], black-box alpha divergence minimisation [14], Rényi divergence VI [24], natural gradient VI [20] and functional variational BNNs [37] all frequently use $\mathcal{Q}_{\text{FFG}}$.

**Mean-Field Variational Inference** Variational inference [2, 19] is an approximate inference method that selects $q_\phi$ by minimising the KL divergence between the approximate and exact posterior [3]. This is equivalent to maximising an evidence lower bound (ELBO): $\mathcal{L}(\phi) = \sum_{n=1}^{N}\mathbb{E}_{q_\phi}\left[\log p(y_n|\mathbf{x}_n, f_\theta)\right] - \text{KL}\left[q_\phi(\theta)||p(\theta)\right]$. Most commonly in BNNs, $q_\phi$ is chosen from $\mathcal{Q}_{\text{FFG}}$. This is known as mean-field variational inference (MFVI).

**Monte Carlo Dropout** MCDO with $\ell_2$ regularisation has been interpreted as VI [10]. Although the MCDO objective is not strictly an ELBO [17], we will sometimes refer to it as such. The variational family, $\mathcal{Q}_{\text{MCDO}}$, is the set of distributions determined by random variables of the form $\widehat{\mathbf{W}} := \mathbf{W}\text{diag}(\boldsymbol{\epsilon})$; where the weights $\mathbf{W}$ are variational parameters and $\epsilon_i \overset{\text{i.i.d}}{\sim} \text{Bern}(1-p)$, with $p$ the dropout probability. Frequently, the first weight matrix $\mathbf{W}_1$ is deterministic (i.e. inputs are not dropped out) — we analyse this case in the main body and use $\mathcal{Q}_{\text{MCDO}}$ to refer to this family. There are fundamentally different considerations when $\mathbf{W}_1$ is also stochastic, addressed in appendix E.

## 2.2 BNN Priors and References for the Exact Posterior Predictive

In this paper, we examine how closely approximate BNN predictive distributions resemble exact inference. To make this comparison, a choice of BNN prior must be made. Common practice is to choose an independent $\mathcal{N}(0, 1)$ prior for all parameters, regardless of the size of the network. However, such priors are known to lead to extremely large variance in function space for wide or deep networks [30]. For example, choosing such a prior for a 4-hidden layer BNN with 50 neurons in each layer leads to a prior standard deviation of $\sim 10^3$ in function space at the origin. This is orders of magnitude too large for normalised data. It is conceivable that one may combine an unreasonable prior with poor approximate inference to obtain practically useful uncertainty estimates that bear little relation to the exact Bayesian predictive — we do not consider this case. Instead, we focus our study on the quality of approximate inference in models with moderate prior variances in function space.

There is a body of literature on BNN priors [30, 28, 35, 22] which shows how to select prior weight variances that lead to reasonable prior variances in function space, even as the width of the hidden layers tends to infinity. For a layer with $N_{\text{in}}$ inputs, we choose independent $\mathcal{N}(0, \sigma_w^2/N_{\text{in}})$ priors for the weights, with $\sigma_w^2$ a constant. For regression with a Gaussian likelihood, as the width tends to infinity, both the prior and posterior of such a BNN converges to a Gaussian process (GP) [18, 30, 28]. It has been shown that even moderately wide BNNs closely resemble their corresponding infinite-width GP counterparts [28]. In this work, we use exact inference in the corresponding infinite-width limit GP and also 'gold-standard' Hamiltonian Monte Carlo (HMC) [31, 16] as references for the exact posterior.

## 3 Single-Hidden Layer Neural Networks

In this section, we prove that for single-hidden layer (1HL) ReLU BNNs, $\mathcal{Q}_{\text{FFG}}$ and $\mathcal{Q}_{\text{MCDO}}$ are not expressive enough to satisfy **Criterion 1**. We identify limitations on the variance in function-space, $\mathbb{V}[f(\mathbf{x})]$, implied by these families. We show empirically that the exact posterior does not have these restrictions, implying that approximate inference does not qualitatively resemble the posterior.

**Theorem 1** (Factorised Gaussian). *Consider any 1HL fully-connected ReLU NN $f : \mathbb{R}^D \to \mathbb{R}$. Let $x_d$ denote the $d^{th}$ element of the input vector $\mathbf{x}$. Assume a fully factorised Gaussian distribution over the parameters. Consider any points $\mathbf{p}, \mathbf{q}, \mathbf{r} \in \mathbb{R}^D$ such that $\mathbf{r} \in \overrightarrow{\mathbf{pq}}$ and either: i. $\overrightarrow{\mathbf{pq}}$ contains $\mathbf{0}$ and $\mathbf{r}$ is closer to $\mathbf{0}$ than both $\mathbf{p}$ and $\mathbf{q}$, or ii. $\overrightarrow{\mathbf{pq}}$ is orthogonal to and intersects the plane $x_d = 0$, and $\mathbf{r}$ is closer to the plane $x_d = 0$ than both $\mathbf{p}$ and $\mathbf{q}$. Then $\mathbb{V}[f(\mathbf{r})] \leq \mathbb{V}[f(\mathbf{p})] + \mathbb{V}[f(\mathbf{q})]$.*

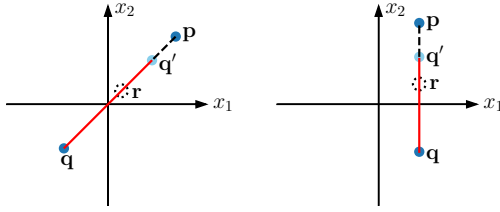

Figure 1: Illustration of the bounded regions in theorem 1, showing the input domain of a 1HL mean-field Gaussian BNN, for the case $\mathbf{x} \in \mathbb{R}^2$. Left (resp. Right): For any two points $\mathbf{p}$ and $\mathbf{q}$ such that the line joining them crosses the origin (resp. is orthogonal to and intersects a plane $x_d = 0$), the output variance at any point $\mathbf{r}$ on the solid red portion of the line is upper bounded by $\mathbb{V}[f(\mathbf{p})] + \mathbb{V}[f(\mathbf{q})]$, illustrating condition (i) (resp. condition (ii)) of theorem 1. The bounded region extends from $\mathbf{q} = (q_1, q_2)$ to $\mathbf{q}'$, where $\mathbf{q}' = (-q_1, -q_2)$ (Left), or $\mathbf{q}' = (q_1, -q_2)$ (Right).

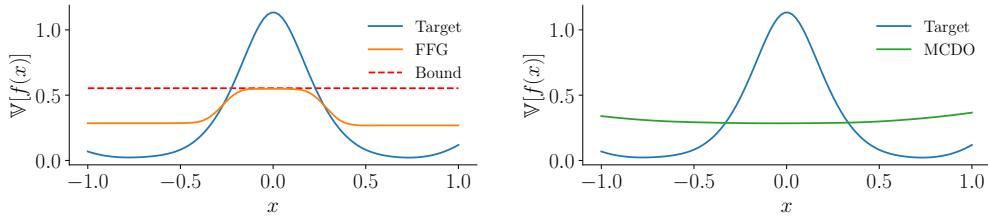

Figure 2: Results of *directly minimising the squared error in function space* between $\mathbb{V}[f(x)]$ (for a single-hidden layer NN) and a target variance function. Left: FFG distribution, Right: MCDO distribution. The bound for FFG distributions (red) applies on $[-1, 1]$ with $\mathbf{p} = -1, \mathbf{q} = 1$. The MCDO variance function is convex, and almost constant. The FFG and MCDO variance functions underestimate the target near the origin and overestimate it away from the origin.

Theorem 1 states that there are line segments (illustrated in figure 1) in input space such that the predictive variance on the line is bounded by the sum of the variance at the endpoints. Analogous but weaker bounds on higher dimensional sets in input space enclosed by these lines can be obtained as a corollary (see appendix B). Theorem 1 applies to any method using $\mathcal{Q}_{\text{FFG}}$, as listed in section 2.1. Although theorem 1 only bounds certain lines in input space, in appendix A we provide figures empirically showing that lines joining random points in input space suffer from similar behaviour. We provide similar results for MC dropout:

**Theorem 2** (MC dropout). *Consider the same network architecture as in theorem 1. Assume an MC dropout distribution over the parameters, with inputs not dropped out. Then $\mathbb{V}[f(\mathbf{x})]$ is convex in $\mathbf{x}$.*

**Remark 1.** *In appendix E, we consider MC dropout with the inputs also dropped out. We prove that the variance at the origin is bounded by the maximum of the variance at any set of points containing the origin in their convex hull. This also applies to variational Gaussian dropout [21]. In the main body, we assume inputs are not dropped out.*

Theorem 2 implies the predictive variance on any line segment in input space is bounded by the maximum of the variance at its endpoints. Full proofs of theorems 1 and 2 are in appendix B. Theorems 1 and 2 show that there are simple cases where 1HL approximate BNNs using $\mathcal{Q}_{\text{FFG}}$ and $\mathcal{Q}_{\text{MCDO}}$ cannot represent *in-between uncertainty*: i.e., increased uncertainty in between well separated regions of low uncertainty. As theorems 1 and 2 depend only on the approximating family, this cannot be fixed by improving the optimiser, regulariser or prior. Figure 2 shows a numerical verification of theorems 1 and 2. Since we are concerned with whether there are *any* distributions that show in-between uncertainty, we do not maximise the ELBO in this experiment (we consider ELBO maximisation in sections 3.2 and 4.2). Instead, we train 1HL networks of width 50 with $\mathcal{Q}_{\text{FFG}}$ and $\mathcal{Q}_{\text{MCDO}}$ distributions to *directly* minimise the squared error between $\mathbb{V}[f(x)]$ and a pre-specified target variance function displaying in-between uncertainty. Full details are given in appendix E.3. Although theorems 1 and 2 apply only to 1HL BNNs, 1HL BNN regression tasks are a very common benchmark in the BNN literature [29, 38, 11, 13, 37], and have been used to assess different inference methods.

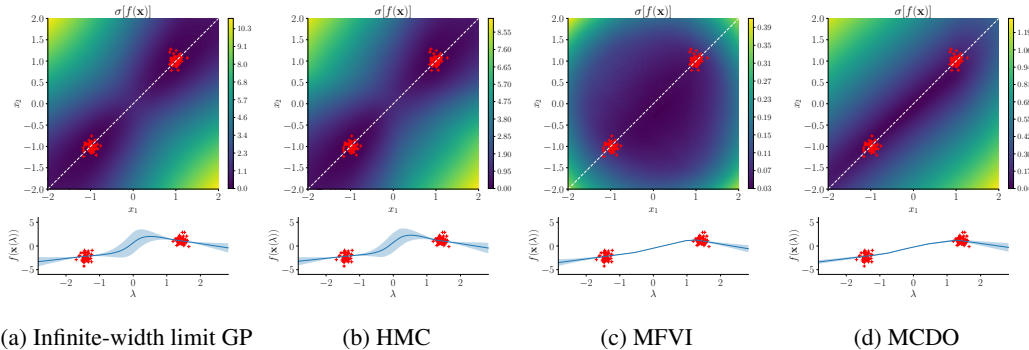

| (a) Infinite-width limit GP | (b) HMC | (c) MFVI | (d) MCDO |

Figure 3: Regression on a 2D synthetic dataset (red crosses). The colour plots show the standard deviation of the output, $\sigma[f(\mathbf{x})]$, in 2D input space. The plots beneath show the mean with 2-standard deviation bars along the dashed white line (parameterised by $\lambda$). MFVI and MCDO are overconfident for $\lambda \in [-1, 1]$. Theorems 1 and 2 explain this: given the uncertainty is near zero at the data, there is *no* setting of the variational parameters that has variance much greater than zero in the line segment between them.

### 3.1 Intuition for Theorems 1 and 2

We now provide intuition for the proofs of theorems 1 and 2. Let $\theta_{\mathrm{in}}$ be the parameters in the first layer. By the law of total variance, $\mathbb{V}[f(\mathbf{x})] = \mathbb{E}[\mathbb{V}[f(\mathbf{x})|\theta_{\mathrm{in}}]] + \mathbb{V}[\mathbb{E}[f(\mathbf{x})|\theta_{\mathrm{in}}]]$. For $\mathcal{Q}_{\mathrm{MCDO}}$ the second term is 0 as $\theta_{\mathrm{in}}$ is deterministic. Hence to prove theorem 2, it suffices to show the first term is convex. We have:

$$\mathbb{V}[f(\mathbf{x})|\theta_{\mathrm{in}}] = \mathbb{V}\left[\sum_{i=1}^{I} w_i \psi(a_i(\mathbf{x}; \theta_{\mathrm{in}})) + b \bigg| \theta_{\mathrm{in}}\right] = \sum_{i=1}^{I} \mathbb{V}[w_i]\psi(a_i(\mathbf{x}; \theta_{\mathrm{in}}))^2 + \mathbb{V}[b], \qquad (2)$$

where $\{w_i\}_{i=1}^{I}$ and $b$ are the output weights and bias, $\psi(a) = \max(0, a)$, and $a_i(\mathbf{x}; \theta_{\mathrm{in}})$ is the activation of the $i^{\mathrm{th}}$ neuron. Since $a_i(\mathbf{x}; \theta_{\mathrm{in}})$ is affine in $\mathbf{x}$, $\psi(a_i(\mathbf{x}; \theta_{\mathrm{in}}))^2$ is a 'rectified quadratic' in $\mathbf{x}$ and therefore convex. This proves theorem 2. The same argument also applies to show that $\mathbb{V}[f(\mathbf{x})|\theta_{\mathrm{in}}]$ is convex for $\mathcal{Q}_{\mathrm{FFG}}$. To arrive at equation (2), we used that for $\mathcal{Q}_{\mathrm{FFG}}$ and $\mathcal{Q}_{\mathrm{MCDO}}$, the output weights of each neuron are independent. Correlations between the weights could introduce negative covariance terms, leading to non-convex behaviour. Thus we see how *weight-space* factorisation assumptions can lead to *function-space* restrictions on the predictive uncertainty.

To complete the proof of theorem 1, we need to analyse $\mathbb{V}[\mathbb{E}[f(\mathbf{x})|\theta_{\mathrm{in}}]]$. Because of the factorisation assumptions on the weights in the first layer, this term is a linear combination of the variances of each activation function. While these variances are not convex, in appendix B we show they satisfy restrictive conditions that imply bounds on arbitrary positive linear combinations of these functions.

### 3.2 Empirical Tests of Approximate Inference in Single-Hidden Layer BNNs

It is not immediately apparent that theorems 1 and 2 are problematic from the perspective of Bayesian inference. For example, even exact inference in a Bayesian linear regression model results in a convex predictive variance function. Here we provide strong evidence that, in contrast, the modelling assumptions of 1HL BNNs lead to *exact* posteriors that *do* show in-between uncertainty. Theorems 1 and 2 thus imply that it is *approximate* inference with $\mathcal{Q}_{\mathrm{FFG}}$ or $\mathcal{Q}_{\mathrm{MCDO}}$ that fails to reflect this intuitively desirable property of the exact predictive, violating **Criterion 1**.

Figure 3 compares the predictive distributions obtained from MFVI and MCDO (here we optimise the ELBO for MFVI and the standard MCDO objective, in contrast with figure 2 — see appendix F for experimental details) with HMC and the limiting GP on a regression dataset consisting of two clusters of covariates. We use 1HL BNNs with 50 hidden units and ReLU activations. The HMC and limiting GP posteriors are almost indistinguishable, suggesting they both resemble the exact predictive. For these methods $\mathbb{V}[f(\mathbf{x})]$ is markedly larger near the origin than near the data. In contrast, MFVI and MCDO are as confident in between the data as they are near the data. This provides strong evidence that the lack of in-between uncertainty is not a feature of the BNN model or prior, but is caused by approximate inference.

# 4 Deeper Networks

Theorems 1 and 2 pose an important question: is the structural limitation observed in the 1HL case fundamental to $\mathcal{Q}_{\mathrm{FFG}}$ and $\mathcal{Q}_{\mathrm{MCDO}}$ even in deeper networks, or can depth help these approximations satisfy **Criterion 1**? In theorem 3, we provide universality results for the mean and variance functions of approximate BNNs with at least two hidden layers using $\mathcal{Q}_{\mathrm{FFG}}$ and $\mathcal{Q}_{\mathrm{MCDO}}$. As the predictive mean and variance often determine the performance of BNNs in regression applications, this provides theoretical evidence that approximate inference in *deep* BNNs satisfies **Criterion 1**.

**Theorem 3** (Deeper networks). *Let $g$ be any continuous function on a compact set $A \subset \mathbb{R}^D$, and $h$ be any continuous, non-negative function on $A$. For any $\epsilon > 0$, for both $\mathcal{Q}_{\mathrm{FFG}}$ and $\mathcal{Q}_{\mathrm{MCDO}}$ there exists a 2HL ReLU BNN such that $\sup_{\mathbf{x} \in A} |\mathbb{E}[f(\mathbf{x})] - g(\mathbf{x})| < \epsilon$ and $\sup_{\mathbf{x} \in A} |\mathbb{V}[f(\mathbf{x})] - h(\mathbf{x})| < \epsilon$.*

**Remark 2.** *If MC dropout is used with inputs also dropped out, the analogous statement to theorem 3 is false. In appendix E, we provide a counterexample that holds for arbitrarily deep networks and shows that if this is the case, $\mathbb{V}[f]$ cannot be made small at two points $\mathbf{x}_1, \mathbf{x}_2$ which have significantly different values of $\mathbb{E}[f(\mathbf{x}_1)]$ and $\mathbb{E}[f(\mathbf{x}_2)]$.*

Figure 4 shows the result of directly minimising the squared error between the network output mean and variance and a given target mean and variance function, using the same method and architecture as with the 1HL network in figure 2. In contrast to figure 2, the variances of both $\mathcal{Q}_{\mathrm{FFG}}$ and $\mathcal{Q}_{\mathrm{MCDO}}$ are able to fit the target.

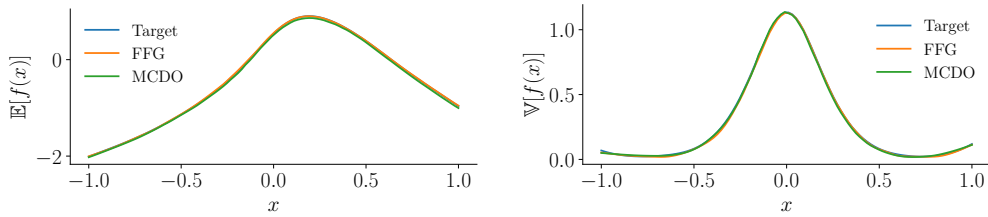

Figure 4: Results of minimising the squared error in function space between $\mathbb{E}[f(x)]$ and a target mean function (left), and between $\mathbb{V}[f(x)]$ and a target variance function (right), for a 2-hidden layer BNN with FFG and MCDO distributions.

While theorem 3 gives some cause for optimism for approximating family methods with deep BNNs, it only shows that the mean and variance of marginal distributions of the output are universal (it does not tell us about higher moments or covariances between outputs). Additionally, it does not say whether good distributions will actually be *found* by an optimiser when maximising the ELBO, i.e it does not address **Criterion 2**.

## 4.1 Proof Sketch of Theorem 3

To prove theorem 3 for $\mathcal{Q}_{\mathrm{FFG}}$, we provide a construction that relies on the universal approximation theorem for deterministic NNs [23]. Consider a 2HL NN whose second hidden layer has two neurons, with activations $a_1, a_2$. Let $w_1, w_2$ denote the weights connecting $a_1, a_2$ to the output, and $b$ denote the output bias, such that the output $f(\mathbf{x}) = w_1 \psi(a_1) + w_2 \psi(a_2) + b$. In this construction, $a_1$ will be used to control the mean, and $a_2$ the variance, of the BNN output. By setting the variances in the first two linear layers to be sufficiently small, we can consider $a_1$ and $a_2$ to be essentially deterministic functions of $\mathbf{x}$. By the universal approximation theorem, $a_1$ and $a_2$ can approximate any continuous functions. Choose $a_1 \approx g(\mathbf{x}) - \min_{\mathbf{x}' \in A} g(\mathbf{x}')$ and $a_2 \approx \sqrt{h(\mathbf{x})}$. Choose $\mathbb{E}[b] = \min_{\mathbf{x}' \in A} g(\mathbf{x}')$, $\mathbb{V}[b] \approx 0$; $\mathbb{E}[w_1] = 1$, $\mathbb{V}[w_1] \approx 0$; and $\mathbb{E}[w_2] = 0$, $\mathbb{V}[w_2] = 1$. By linearity of expectation, the factorisation assumptions, and $a_1, a_2 \geq 0$:

$$\mathbb{E}[f(\mathbf{x})] = \mathbb{E}[w_1 \psi(a_1) + w_2 \psi(a_2) + b] = \mathbb{E}[w_1]\mathbb{E}[\psi(a_1)] + \mathbb{E}[w_2]\mathbb{E}[\psi(a_2)] + \mathbb{E}[b]$$
$$\approx g(\mathbf{x}) - \min_{\mathbf{x}' \in A} g(\mathbf{x}') + \min_{\mathbf{x}' \in A} g(\mathbf{x}') = g(\mathbf{x}),$$

as desired. By the law of total variance, the variance of the network output is

$$\mathbb{V}[f(\mathbf{x})] = \mathbb{E}[\mathbb{V}[f(\mathbf{x})|a_1, a_2]] + \mathbb{V}[\mathbb{E}[f(\mathbf{x})|a_1, a_2]] \approx \mathbb{E}[\mathbb{V}[f(\mathbf{x})|a_1, a_2]] \approx \mathbb{E}[\psi(a_2)^2] + \mathbb{V}[b] \approx h(\mathbf{x}),$$

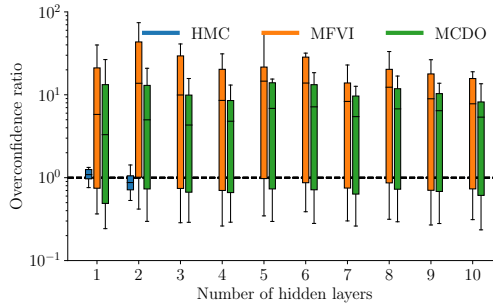

Figure 5: Box and whisker plots of the overconfidence ratios of HMC, MFVI and MCDO relative to exact inference in the corresponding infinite-width limit GP. The whiskers show the smallest and largest overconfidence ratios computed, and the the box extends from the lower to upper quartile values of the overconfidence ratios, with a line at the median. HMC is only run for 1 and 2 hidden layers due to difficulty ensuring convergence in larger models. We fix the BNN width to 50.

where we used that $w_1, b$ are essentially deterministic and $\mathbb{V}[\mathbb{E}[f(\mathbf{x})|a_1, a_2]] \approx 0$ since $a_1, a_2$ are essentially deterministic. Also, we have that $\psi(a_2) \approx a_2$ since $a_2 \approx \sqrt{h(\mathbf{x})} \geq 0$. The approximations come from the standard universal function approximation theorem, and the variances of weights not being set exactly to 0 so that we remain in $\mathcal{Q}_{\mathrm{FFG}}$. A rigorous proof, along with a proof for $\mathcal{Q}_{\mathrm{MCDO}}$ with any dropout rate $p \in (0,1)$, is given in appendix D. The proof for $\mathcal{Q}_{\mathrm{MCDO}}$ uses a similar strategy, but is more involved as we cannot set individual weights to be essentially deterministic.

## 4.2 Empirical Tests of Approximate Inference in Deep BNNs

We now consider empirically whether the distributions found by optimising the ELBO with these families resemble the exact predictive distribution (**Criterion 2**). To do this, we define the 'overconfidence ratio' at an input $\mathbf{x}$ as $\gamma(\mathbf{x}) = (\mathbb{V}_{\mathrm{GP}}[f(\mathbf{x})]/\mathbb{V}_{q_\phi}[f(\mathbf{x})])^{1/2}$, where $\mathbb{V}_{\mathrm{GP}}$ is the predictive variance of exact inference in the infinite-width BNN. We then compute $\gamma(\mathbf{x})$ at 300 points $\{\mathbf{x}_n\}_{n=1}^{300}$ evenly spaced along the dashed white line joining the data clusters in figure 3, i.e., from $\mathbf{x} = (-1.2, -1.2)$ to $\mathbf{x} = (1.2, 1.2)$. We then create boxplots of the values $\{\gamma(\mathbf{x}_n)\}_{n=1}^{300}$ for varying BNN depths, shown in figure 5. Accurate inference should lead to similar uncertainty estimates to the limiting GP, i.e. the boxplot should be tightly centered around 1 (dashed line). For the 1HL and 2HL BNNs, the GP and HMC agree closely, suggesting both resemble the exact predictive. In contrast, MFVI and MCDO are often an order of magnitude overconfident ($\gamma(\mathbf{x}) > 1$) *between* the data clusters (upper tail of the boxplot) and somewhat underconfident ($\gamma(\mathbf{x}) < 1$) *at* the data clusters (lower tail of the boxplot). Increased depth does not alleviate this behaviour. See appendix F for experimental details and figures demonstrating this for different priors. In addition, in appendix A, we plot the uncertainty on line segments in between *random* clusters of data in a 5-dimensional input space, with similar results, showing that this phenomenon is not specific to the dataset from figure 3.

In light of theorem 3, it is perhaps surprising that VI fails to capture important properties of the predictive with deep networks. In appendix G we initialise the variational parameters such that the approximate predictive has mean and variance functions that closely match a reference predictive that exhibits in-between uncertainty. This is done by directly minimising the squared loss between the BNN mean and variance functions and the references. We find that proceeding to optimise the ELBO from this initialisation *still* leads to a lack of in-between uncertainty. This suggests that the objective function is at least partially at fault for the mismatch between the approximate and exact posteriors.

## 5 Case Study: Active Learning with BNNs

We now consider the impact of the pathologies described in sections 3 and 4 on active learning [36] on a real-world dataset, where the task is to use uncertainty information to intelligently select which points to label. Active learning with approximate BNNs has been considered in previous works, often showing improvements over random selection of datapoints [13, 12]. However, in cases when active

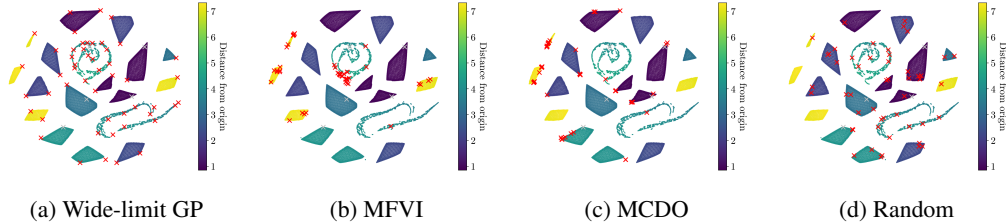

| (a) Wide-limit GP | (b) MFVI | (c) MCDO | (d) Random |

Figure 6: Points chosen during active learning in the 1HL case. Colours denote distance from the origin in 14-dimensional input space. Grey crosses (✕) denote the five points randomly chosen as an initial training set. Red crosses (✕) denote the 50 points selected by active learning. Both MFVI and MCDO entirely miss some clusters which are nearer the origin, and oversample certain clusters which are far from the origin, as might be expected of methods that struggle to represent in-between uncertainty. In contrast, the limiting GP samples the 'corners' of each cluster, without missing any entirely. Note that t-SNE does not preserve relative positions, so that clusters near the origin may appear on the 'outside' of the t-SNE plot.

learning fails, common metrics such as RMSE are insufficient to diagnose the causes. In particular, it is difficult to attribute the failure to the model or to poor approximate inference. In this section, we specifically analyse a dataset where we have observed active learning with approximate BNNs to fail — the Naval regression dataset [6], which is 14-dimensional and consists of 11,934 datapoints. We find via PCA that this dataset has most of its variance along a single direction. It hence may be especially problematic for methods that struggle with in-between uncertainty, as points are more likely to lie roughly in between others.

The main questions we address are: i) Is a lack of in-between uncertainty indicative of pathological behaviour in the 1HL case? In higher dimensional datasets such as Naval, it is not immediately apparent that theorems 1 and 2 are problematic, since the convex hull of the datapoints may have low volume in high dimensions. However, these theorems may be symptomatic of *related* problems with 1HL BNN uncertainty estimates. ii) Given theorem 3, will deeper approximate BNNs usefully reflect BNN modelling assumptions for active learning?

**Experimental Set-up and Results**   We compare MFVI, MCDO and the limiting GP. We do not run HMC as this would take too long to wait for convergence at each iteration of active learning. We normalise the dataset to have zero mean and unit standard deviation in each dimension. 5 datapoints are chosen randomly as an initial active set, with the rest being the pool set. Models are trained on the active set, then the datapoint from the pool set that has the highest predictive variance is added to the active set, following Hernández-Lobato and Adams [13]. We train MFVI and MCDO for 20,000 iterations of ADAM at each step of active learning. This process is repeated 50 times. Table 1 shows the RMSE of each model on a held-out test set after this process, compared to a baseline where points are chosen randomly. Full details are in appendix H.1. Active learning significantly reduces RMSE for the GP compared to random selection, often by more than a factor of three. However it *increases* RMSE for 1HL MFVI and MCDO, and either increases it or does not significantly decrease it for deeper networks. The one exception is 3HL MCDO, where active performs about 10% better than random, which is still far less than the factor of three improvement suggested by exact inference in the infinite-width BNN.

**Discussion**   In figure 6 we visualise the dataset using t-SNE [39]. The covariates of Naval are clustered, with points in the same cluster roughly the same distance from the origin. Since the

Table 1: Test RMSEs ($\pm$ 1 standard error) after the $50^{\text{th}}$ iteration of active learning, averaged over 20 random seeds. As the data is normalised, a method that predicts 0 will have an RMSE near 1.

|             | 1 HL          | 2 HL          | 3 HL          | 4 HL          |
|-------------|---------------|---------------|---------------|---------------|
| GP Active   | $0.04 \pm 0.00$ | $0.04 \pm 0.00$ | $0.04 \pm 0.00$ | $0.05 \pm 0.00$ |
| GP Random   | $0.12 \pm 0.01$ | $0.13 \pm 0.01$ | $0.15 \pm 0.01$ | $0.16 \pm 0.01$ |
| MFVI Active | $0.94 \pm 0.11$ | $0.46 \pm 0.04$ | $0.35 \pm 0.03$ | $0.31 \pm 0.02$ |
| MFVI Random | $0.15 \pm 0.01$ | $0.23 \pm 0.01$ | $0.28 \pm 0.01$ | $0.32 \pm 0.01$ |
| MCDO Active | $0.69 \pm 0.04$ | $0.36 \pm 0.02$ | $0.38 \pm 0.02$ | $0.45 \pm 0.02$ |
| MCDO Random | $0.22 \pm 0.01$ | $0.35 \pm 0.01$ | $0.43 \pm 0.01$ | $0.47 \pm 0.02$ |

dataset is mean-centred, points closer to the origin are in a sense 'in between' others. We see that although the GP chooses points from every cluster during active learning, MFVI fails to select any points from many clusters — including all the clusters closest to the origin. It ignores points in the 'inside' and oversamples points on the 'outside', leading to a selection strategy worse than random. This behaviour is consistent with theorem 1. MCDO also fails to sample from many clusters; in appendix H.2 we show this is because it fails to reduce its uncertainty at clusters it has already heavily sampled. Interestingly, it sometimes chooses from clusters near the origin, even though its variance function is provably convex. This may be because the minimum of the variance function for MCDO is not centred at the origin, or because the variance has the shape of an elongated valley. In contrast, the GP seems to select the 'corners' of each cluster, which is intuitively efficient. The success of the infinite-width GP provides evidence that this BNN model combined with exact inference has desirable inductive biases for this task; it is *approximate* inference that has caused active learning to fail. In deeper networks, theorem 3 gives hope that the BNN predictive variance may be useful for active learning. While we find the problems are indeed less severe than in the 1HL case, MFVI still oversamples points away from the origin compared to those near the origin (see appendix H.2).

## 6  Related Work

Concerns have been raised about the suitability of $\mathcal{Q}_{FFG}$ since early work on BNNs. MacKay [27, Figure 1] noted that a full-covariance Gaussian family was needed to obtain predictions with increased uncertainty away from data with the Laplace approximation, although no detailed explanation was provided. The desire to go beyond $\mathcal{Q}_{FFG}$ has motivated a great deal of research into more flexible approximating families [1, 26, 34]. However, to our knowledge, theorem 1 is the first theoretical result showing that $\mathcal{Q}_{FFG}$ can have a pathologically restrictive effect on BNN predictive uncertainties.

Recently, Farquhar et al. [8] argued that MFVI becomes a less restrictive approximation as depth increases in BNNs. However, they use different criteria to assess the quality of approximate inference. Farquhar et al. [8] use performance on classification benchmarks such as ImageNet and also the KL-divergence between certain Gaussian approximations to HMC samples in weight space to evaluate inference. In contrast, we investigate the resemblance between the function-space predictive distributions of the approximate and exact posteriors with the same prior, and focus on separating the effects of modelling and approximate inference. Additionally, we do not consider posterior tempering, and we use a different scaling for our BNN priors (see section 2.2). Recently, Wenzel et al. [41] also performed a study of the quality of approximate inference in deep BNNs. They focused on stochastic gradient Markov Chain Monte Carlo (SGMCMC) [40, 5, 42] in deep convolutional networks, and concluded that SGMCMC is accurate enough for inference, suggesting that the prior is at fault for poor performance. In contrast, we provide theoretical and empirical evidence for a specific pathology in VI in lower-dimensional regression problems, and demonstrate cases where BNN priors *do* encode useful inductive biases which are subsequently lost by approximate inference.

Osband et al. [32] note that the MCDO predictive distribution is invariant to duplicates of the data, and in the linear case predictive uncertainty does not decrease as dataset size increases (if the dropout rate and regulariser are fixed[2]). Theorem 2 shows that in the non-linear 1HL case, predictive uncertainty in the MCDO posterior is fundamentally restricted even for datasets without repeated entries.

## 7  Conclusions

Principled approximate Bayesian inference involves defining a reasonable model, then finding an approximate posterior that retains the relevant properties of the exact posterior. We have shown that for BNNs, in-between uncertainty is a feature of the predictive that is often lost by variational inference. Although this is of greatest relevance for lower-dimensional regression tasks, the fact that MFVI and MCDO often fail these simple sanity checks indicates that these methods might generally have predictive distributions which are qualitatively different from the exact predictive. While BNNs have previously been shown to provide uncertainty estimates that are useful for a range of tasks, it remains an open question as to what extent this is attributable to a resemblance between the approximate and exact predictive posteriors.

## Broader Impact

Bayesian approaches to deep learning problems are often proposed in situations where uncertainty estimation is critical. Often the justification given for this approach is the probabilistic framework of Bayesian inference. However, in cases where approximations are made, the quality of these approximations should also be taken into account. Our work illustrates that the uncertainty estimates given by approximate inference with commonly used algorithms may not qualitatively resemble the uncertainty estimates implied by Bayesian modelling assumptions. This may possibly have adverse consequences if Bayesian neural networks are used in safety-critical applications. Our work motivates a careful consideration of these situations.

## Acknowledgments and Disclosure of Funding

We thank Wessel Bruinsma for the proof of lemma 5, and José Miguel Hernández-Lobato, Ross Clarke and Sebastian W. Ober for helpful discussions. AYKF gratefully acknowledges funding from the Trinity Hall Research Studentship and the George and Lilian Schiff Foundation. DRB gratefully acknowledges funding from the Herchel Smith Fellowship through Williams College, as well as the Qualcomm Innovation Fellowship. RET is supported by Google, Amazon, ARM, Improbable, EPSRC grants EP/M0269571 and EP/L000776/1, and the UKRI Centre for Doctoral Training in the Application of Artificial Intelligence to the study of Environmental Risks (AI4ER).

## Footnotes

[2]Note that for a fixed prior, the 'KL condition' [10, Section 3.2.3] requires the $\ell_2$ regularisation constant to decrease with increasing dataset size.

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
