[Supplementary Material]

# Supplement: On the Expressiveness of Approximate Inference in Bayesian Neural Networks

**Andrew Y. K. Foong**[*]
University of Cambridge
ykf21@cam.ac.uk

**David R. Burt**[*]
University of Cambridge
drb62@cam.ac.uk

**Yingzhen Li**
Microsoft Research
Yingzhen.Li@microsoft.com

**Richard E. Turner**
University of Cambridge
Microsoft Research
ret26@cam.ac.uk

## A  In-between Uncertainty in Other Regions of Input Space

In this appendix, we show plots generated by placing two Gaussian clusters of data with centers randomly chosen on the sphere of radius $\sqrt{D}$, where $D = 5$ denotes input dimension. We generate synthetic data by sampling from a wide-limit Gaussian process. For each plot, we show the predictive mean and 2 standard deviations along the line segment in input space joining the centres of these two clusters. For all plots, we choose $\sigma_w = \sqrt{2}$, $\sigma_b = 1$, networks of width 50 and dropout probability of $p = 0.05$ for MCDO. We set the observation noise standard deviation to 0.01, which is the ground truth value used to generate the synthetic data. The initialisation of MFVI and MCDO is the same as discussed in appendix F.

In the 1HL case, Theorem 2 implies that MCDO's predictive variance will be convex along any line, including the line plotted. In contrast, theorem 1 only applies to certain lines in input space, and does not bound the variance in these cases. However, we suspect that theorem 1 is indicative of a lack of in-between uncertainty in more general cases. Figure 6 shows that although that exact inference with the GP with the limiting BNN prior exhibits in-between uncertainty, this is lost by both MFVI and MCDO, even on general lines in input space. MFVI and MCDO are often *more* confident in between the data clusters than at the data clusters.

We next run the same experiment but with 3HL BNNs and their limiting GP. In this case theorem 3 implies that for sufficiently wide BNNs, there exist variational parameters that can approximate *any* predictive mean and standard deviation. However, in figure 7 we see that compared to exact inference in the limiting GP, MFVI and MCDO both underestimate in-between uncertainty — or sometimes show as large uncertainty *at* the data as *in between* the data.

## B  General Statements and Proofs of Theorems 1 and 2

In section 3 we stated simplified versions of bounds concerning the variance of single-hidden layer networks with certain approximating families. The two main results we prove in this section are the following generalisations of theorems 1 and 2 respectively:

**Theorem 1.** *Consider a single-hidden layer ReLU neural network mapping from $\mathbb{R}^D \rightarrow \mathbb{R}^K$ with $I \in \mathbb{N}$ hidden units. The corresponding mapping is given by $f^{(k)}(\mathbf{x}) = \sum_{i=1}^{I} w_{k,i} \psi \left( \sum_{d=1}^{D} u_{i,d} x_d + v_i \right) + b_k$ for $1 \leq k \leq K$, where $\psi(a) = \max(0, a)$. Suppose we*

---

[*]Equal contribution.

Figure 6: Mean and 2 standard deviation bars of the predictive distribution on lines joining random clusters of data, for 1HL BNNs. Each row represents the same random dataset. We also plot the projection of the 5-dimensional data onto this line segment as its $\lambda$-value, along with the output value of the data. Note that the data appears noisy, but this is due to the projection onto a lower-dimensional space.

*have a distribution over network parameters with density of the form:*

$$q(\mathbf{W}, \mathbf{b}, \mathbf{U}, \mathbf{v}) = \prod_{i=1}^{I} q_i(\mathbf{w}_i | \mathbf{U}, \mathbf{v}) q(\mathbf{b} | \mathbf{U}, \mathbf{v}) \prod_{i=1}^{I} \prod_{d=1}^{D} \mathcal{N}(u_{i,d}; \mu_{u_{i,d}}, \sigma^2_{u_{i,d}}) \prod_{i=1}^{I} \mathcal{N}(v_i; \mu_{v_i}, \sigma^2_{v_i}), \quad (1)$$

*where $\mathbf{w}_i = \{w_{k,i}\}_{k=1}^{K}$ are the weights out of neuron $i$ and $\mathbf{b} = \{b_k\}_{k=1}^{K}$ are the output biases, and $q_i(\mathbf{w}_i | \mathbf{U}, \mathbf{v})$ and $q(\mathbf{b} | \mathbf{U}, \mathbf{v})$ are arbitrary probability densities with finite first two moments. Consider a line in $\mathbb{R}^D$ parameterised by $\mathbf{x}(\lambda)_d = \gamma_d \lambda + c_d$ for $\lambda \in \mathbb{R}$ such that $\gamma_d c_d = 0$ for $1 \leq d \leq D$. Then for any $\lambda_1 \leq 0 \leq \lambda_2$, and any $\lambda_*$ such that $|\lambda_*| \leq \min(|\lambda_1|, |\lambda_2|)$,*

$$\mathbb{V}[f^{(k)}(\mathbf{x}(\lambda_*))] \leq \mathbb{V}[f^{(k)}(\mathbf{x}(\lambda_1))] + \mathbb{V}[f^{(k)}(\mathbf{x}(\lambda_2))] \quad \text{for} \quad 1 \leq k \leq K. \quad (2)$$

We now briefly show how the statement of theorem 1 in the main text can be deduced from this more general version. The fully factorised Gaussian family $\mathcal{Q}_{\mathrm{FFG}}$ is of the form in equation 1. It remains to show that both conditions $i.$ and $ii.$ imply that $\gamma_d c_d = 0$. Consider any line intersecting the origin (i.e. satisfying condition i)). Such a line can be written in the form $\mathbf{x}(\lambda)_d = \gamma_d \lambda$ by choosing the origin to correspond to $\lambda = 0$. As $c_d = 0$ for all $d$, $\gamma_d c_d = 0$ for all $d$. In theorem 1 $\mathbf{p} = \mathbf{x}(\lambda_1)$ and $\mathbf{q} = \mathbf{x}(\lambda_2)$ are on opposite sides of the origin, hence the signs of $\lambda_1$ and $\lambda_2$ are opposite. Finally, the condition that $\mathbf{r} = \mathbf{x}(\lambda_*)$ is closer to the origin than both $\mathbf{p}$ and $\mathbf{q}$ is exactly that $|\lambda_*| \leq \min(|\lambda_1|, |\lambda_2|)$.

In order to verify condition ii), note that any line orthogonal to a hyperplane $x_{d'} = 0$ can be parameterised as $\mathbf{x}(\lambda)_d = \gamma_d \lambda + c_d$, where $\gamma_d = 0$ for $d \neq d'$ and $c_{d'} = 0$. Hence $\gamma_d c_d = 0$ for all $d$. The condition that the line segment $\overrightarrow{\mathbf{pq}}$ intersects the plane, with $\mathbf{p} = \mathbf{x}(\lambda_1)$ and $\mathbf{q} = \mathbf{x}(\lambda_2)$ is exactly that the signs of $\lambda_1$ and $\lambda_2$ are opposite, and that $|\lambda_*| \leq \min(|\lambda_1|, |\lambda_2|)$.

Figure 7: Same experimental set-up as in figure 6 for the 3HL case.

As a corollary of theorem 1, we can obtain bounds on higher-dimensional objects than lines, such as on hypercubes. For instance, consider the case where $\mathbf{x} \in \mathbb{R}^2$. Let $\mathbf{p}, \mathbf{q}, \mathbf{r}, \mathbf{s}$ be the four corners of a rectangle centered the origin. For any point $\mathbf{a}$ in the rectangle, we can upper bound $\mathbb{V}[f(\mathbf{a})]$ by the sum of the variances at the top and bottom edges of the rectangle. These in turn can be upper bounded by the variances at the corners of the rectangle. Hence we have that for any point $\mathbf{a}$ in the rectangle, $\mathbb{V}[f(\mathbf{a})] \leq \mathbb{V}[f(\mathbf{p})] + \mathbb{V}[f(\mathbf{q})] + \mathbb{V}[f(\mathbf{r})] + \mathbb{V}[f(\mathbf{s})]$. Similarly the variance at any point in a hypercube centered at the origin can be bounded by the sum of the variances on its vertices, and we can obtain tighter bounds on diagonals and faces of the hypercube, by repeatedly applying theorem 1.

**Theorem 2** (MC dropout). *Consider a single-hidden layer ReLU neural network mapping from $\mathbb{R}^D \rightarrow \mathbb{R}^K$ with $I \in \mathbb{N}$ hidden units. The corresponding mapping is given by $f^{(k)}(\mathbf{x}) = \sum_{i=1}^{I} w_{k,i} \psi \left( \sum_{d=1}^{D} u_{i,d} x_d + v_i \right) + b_k$ for $1 \leq k \leq K$, where $\psi(a) = \max(0, a)$. Assume $\mathbf{U}, \mathbf{v}$ are set deterministically and*

$$q(\mathbf{W}, \mathbf{b}) = q(\mathbf{b}) \prod_{i=1}^{I} q_i(\mathbf{w}_i),$$

*where $\mathbf{w}_i = \{w_{k,i}\}_{k=1}^{K}$ are the weights out of neuron $i$, $\mathbf{b} = \{b_k\}_{k=1}^{K}$ are the output biases and $q(\mathbf{b})$ and $q_i(\mathbf{w}_i)$ are arbitrary probability densities with finite first two moments. Then, $\mathbb{V}[f^{(k)}(\mathbf{x})]$ is convex in $\mathbf{x}$ for $1 \leq k \leq K$.*

*Proof.* The theorem follows immediately from lemma 1 since $\mathbf{U}$ and $\mathbf{v}$ are deterministic. $\square$

**Remark 1.** *Theorem 2 applies for any activation function $\psi$ such that $\psi^2$ is convex. This is the only property of $\psi$ used in lemma 1.*

### B.1 Preliminary Lemmas

In order to prove theorems 1 and 2 we first collect a series of preliminary lemmas.

**Lemma 1.** *Assume a distribution for $\mathbf{W}, \mathbf{b}|\mathbf{U}, \mathbf{v}$ with density of the form*

$$q(\mathbf{W}, \mathbf{b}|\mathbf{U}, \mathbf{v}) = q(\mathbf{b}|\mathbf{U}, \mathbf{v}) \prod_i q_i(\mathbf{w}_i|\mathbf{U}, \mathbf{v}).$$

*Then, $\mathbb{V}[f^{(k)}(\mathbf{x})|\mathbf{U}, \mathbf{v}]$ is a convex function of $\mathbf{x}$.*

The proof of lemma 1 is in appendix C.1.

**Lemma 2.** *Consider the variance of a single neuron in the one dimensional case, with activation $a(x) \sim \mathcal{N}(\mu(x), \sigma^2(x))$, $\mu(x) = \mu_u x + \mu_v$ and $\sigma^2(x) = \sigma_u^2 x^2 + \sigma_v^2$. Let*

$$\mathcal{T}_1 = \{f \geq 0 : \forall 0 \leq b < a, f(a) \geq f(-a) \text{ and } f(b) \leq f(a)\}$$

*and*

$$\mathcal{T}_2 = \{f \geq 0 : \forall a < b \leq 0, f(a) \geq f(-a) \text{ and } f(b) \leq f(a)\}.$$

*If $\mu_u \geq 0$, then $\mathbb{V}[\psi(a(x))] \in \mathcal{T}_1$. If $\mu_u \leq 0$, then $\mathbb{V}[\psi(a(x))] \in \mathcal{T}_2$.*

The proof of lemma 2 is in appendix C.2.

**Corollary 1** (Corollary of lemma 2)**.** *Consider a line in $\mathbb{R}^D$ parameterized by $[\mathbf{x}(\lambda)]_d = \gamma_d \lambda + c_d$ for $\lambda \in \mathbb{R}$ such that $\gamma_d c_d = 0$ for $1 \leq d \leq D$. Let $a(\mathbf{x}) := \sum_{d=1}^D u_d x_d + v$ with $\{u_d\}_{d=1}^D$ and $v$ independent and Gaussian distributed. Then, $\mathbb{V}[\psi(a(\mathbf{x}(\lambda)))] \in \mathcal{T}_1 \cup \mathcal{T}_2$ (as a function of $\lambda$).*

*Proof.* The activation $a(\mathbf{x}(\lambda))$ is a linear combination of Gaussian random variables, and is therefore Gaussian distributed. Moreover the mean is linear in $\lambda$. The variance of $a(\mathbf{x}(\lambda))$ is given by:

$$\mathbb{V}[a(\mathbf{x}(\lambda))] = \sum_{d=1}^D \mathbb{V}[u_d](\gamma_d \lambda + c_d)^2 + \mathbb{V}[v]$$

$$= \sum_{d=1}^D \sigma_{u_d}^2 (\gamma_d \lambda + c_d)^2 + \sigma_v^2$$

$$= \lambda^2 \left( \sum_{d=1}^D \sigma_{u_d}^2 \gamma_d^2 \right) + 2\lambda \left( \sum_{d=1}^D \sigma_{u_d}^2 \gamma_d c_d \right) + \left( \sum_{d=1}^D \sigma_{u_d}^2 c_d^2 + \sigma_v^2 \right)$$

$$= \lambda^2 \left( \sum_{d=1}^D \sigma_{u_d}^2 \gamma_d^2 \right) + \left( \sum_{d=1}^D \sigma_{u_d}^2 c_d^2 + \sigma_v^2 \right).$$

Defining $\sigma_{\tilde{u}}^2 = \sum_{d=1}^D \sigma_{u_d}^2 \gamma_d^2$ and $\sigma_{\tilde{v}}^2 = \sum_{d=1}^D \sigma_{u_d}^2 c_d^2 + \sigma_v^2$, the corollary follows from lemma 2. $\quad\square$

**Lemma 3.** *Let $\mathcal{C}$ be the set of convex functions from $\mathbb{R} \to [0, \infty)$. Fix any $a < 0 < b$ and $c$ such that $|c| \leq \min(|a|, |b|)$. Then any function $f$ that can be written as a linear combination of functions in $\mathcal{T}_1 \cup \mathcal{T}_2 \cup \mathcal{C}$ with non-negative weights satisfies, $f(c) \leq f(a) + f(b)$.*

The proof of lemma 3 can be found in appendix C.3.

## B.2 Proof of Theorem 1

Having collected the necessary preliminary lemmas we now prove theorem 1.

*Proof of Theorem 1.* By the law of total variance,

$$\mathbb{V}[f^{(k)}(\mathbf{x})] = \mathbb{E}[\mathbb{V}[f^{(k)}(\mathbf{x})|\mathbf{U}, \mathbf{v}]] + \mathbb{V}[\mathbb{E}[f^{(k)}(\mathbf{x})|\mathbf{U}, \mathbf{v}]].$$

Using lemma 1, $\mathbb{V}[f^{(k)}(\mathbf{x})|\mathbf{U}, \mathbf{v}]$ is convex as a function of $\mathbf{x}$. As the expectation of a convex function is convex, the first term is a convex function of $\mathbf{x}$. For the second term we have

$$\mathbb{E}[f^{(k)}(\mathbf{x})|\mathbf{U}, \mathbf{v}] = \mathbb{E}\left[ \sum_{i=1}^I w_{k,i} \psi(a_i) + b_k \middle| \mathbf{U}, \mathbf{v} \right] = \sum_{i=1}^I \mu_{w_{k,i}} \psi(a_i) + \mu_{b_k},$$

where $\mu_{w_{k,i}} \coloneqq \mathbb{E}[w_{k,i}], \mu_{b_k} \coloneqq \mathbb{E}[b_k]$. In the second line we used linearity of expectation and that conditioned on $(\mathbf{U}, \mathbf{v})$, the $a_i$ are deterministic. Next,

$$\mathbb{V}[\mathbb{E}[f^{(k)}(\mathbf{x})|\mathbf{U}, \mathbf{v}]] = \mathbb{V}\left[\sum_{i=1}^{I} \mu_{w_{k,i}} \psi(a_i) + \mu_{b_k}\right] = \sum_{i=1}^{I} \mu_{w_{k,i}}^2 \mathbb{V}[\psi(a_i)], \tag{3}$$

since the $a_i$ are independent of each other.

Consider a line in $\mathbb{R}^D$ parameterised by $[\mathbf{x}(\lambda)]_d = \gamma_d \lambda + c_d$ for $\lambda \in \mathbb{R}$ such that $\gamma_d c_d = 0$ for $1 \leq d \leq D$.

By corollary 1, $\mathbb{V}[\psi(a_i(\mathbf{x}(\lambda)))] \in \mathcal{T}_1 \cup \mathcal{T}_2$ (as a function of $\lambda$). Since $\mathbb{V}[f^{(k)}(\mathbf{x})|\mathbf{U}, \mathbf{v}]$ is convex as a function of $\mathbf{x}$, it is also convex as a function of $\lambda$. We have written $\mathbb{V}[f^{(k)}(\mathbf{x}(\lambda))]$ in the form assumed in lemma 3, completing the proof. $\qquad\square$

# C  Proof of Lemmas

In this section we prove the preliminary lemmas stated in appendix B.1.

## C.1  Proof of Lemma 1

*Proof.* We assume a distribution for the network weights such that:

$$q(\mathbf{W}, \mathbf{b}|\mathbf{U}, \mathbf{v}) = q(\mathbf{b}|\mathbf{U}, \mathbf{v}) \prod_{i=1}^{I} q_i(\mathbf{w}_i|\mathbf{U}, \mathbf{v}).$$

By this factorisation assumption, the outgoing weights from each neuron are conditionally independent. This means the conditional variance of the output under this distribution can be written

$$\mathbb{V}[f^{(k)}(\mathbf{x})|\mathbf{U}, \mathbf{v}] = \sum_i \mathbb{V}[w_{k,i}|\mathbf{U}, \mathbf{v}] \psi(a_i)^2 + \mathbb{V}[b_k|\mathbf{U}, \mathbf{v}]. \tag{4}$$

with $a_i \coloneqq a_i(\mathbf{x}) = \sum_{d=1}^{D} u_{i,d} x_d + v_i$.

Since $\mathbb{V}[f^{(k)}(\mathbf{x})|\mathbf{U}, \mathbf{v}]$ is a linear combination of the $\psi(a_i)^2$ with non-negative weights (plus a constant), to prove convexity it suffices to show that each $\psi(a_i)^2$ is convex as a function of $\mathbf{x}$. $\psi(a_i)^2$ is convex as a function of $a_i$, since it is 0 for $a_i \leq 0$ and $a_i^2$ for $a_i > 0$. To show that it is convex as a function of $\mathbf{x}$, we write

$$\begin{aligned}
\psi\left(a_i(t\mathbf{x}_1 + (1-t)\mathbf{x}_2)\right)^2 &= \psi\left(\sum_d u_{i,d}\left(t[\mathbf{x}_1]_d + (1-t)[\mathbf{x}_1]_d\right) + v_i\right)^2 \\
&= \psi\left(t\left(\sum_d u_{i,d}[\mathbf{x}_1]_d + v_i\right) + (1-t)\left(\sum_d u_{i,d}[\mathbf{x}_2]_d + v_i\right)\right)^2 \\
&\leq t\psi\left(\sum_d u_{i,d}[\mathbf{x}_1]_d + v_i\right)^2 + (1-t)\psi\left(\sum_d u_{i,d}[\mathbf{x}_2]_d + v_i\right)^2 \\
&= t\psi\left(a_i(\mathbf{x}_1)\right)^2 + (1-t)\psi\left(a_i(\mathbf{x}_2)\right)^2.
\end{aligned}$$

The inequality uses convexity of $\psi(a)$ as a function of $a$. $\qquad\square$

## C.2  Proof of Lemma 2

Throughout, we assume $\sigma_u, \sigma_v$ and $\mu_v$ are fixed and suppress dependence on these parameters. Let $v_{\mu_u}(x) \coloneqq \mathbb{V}[\psi(a(x))]$ where the variance is taken with respect to a distribution with parameter $\mu_u$. Then, $v_{\mu_u}(x) = v_{-\mu_u}(-x)$ since $\mu(x)$ and $\sigma^2(x)$ are unchanged by the transformation $\mu_u, x \to -\mu_u, -x$.

Suppose $v_{\mu_u} \in \mathcal{T}_1$ for $\mu_u > 0$, then for $x \leq 0$,

$$v_{-\mu_u}(x) = v_{\mu_u}(-x) \geq v_{\mu_u}(x) = v_{-\mu_u}(-x),$$

and for $x < y \leq 0$,
$$v_{-\mu_u}(y) = v_{\mu_u}(-y) \leq v_{\mu_u}(-x) = v_{-\mu_u}(x).$$
In words, if $v_{\mu_u} \in \mathcal{T}_1$ then $v_{-\mu_u} \in \mathcal{T}_2$. It therefore suffices to consider the case when $\mu_u \geq 0$.

We first show that if $x \geq 0, v_{\mu_u}(x) \geq v_{\mu_u}(-x)$. Henceforth, we assume $\mu_u \geq 0$ is fixed and suppress it notationally. From Frey and Hinton [3],
$$v(x) = \sigma(x)^2 \alpha(r(x)), \tag{5}$$
Here $r(x) = \mu(x)/\sigma(x)$. We define $h(r) = N(r) + r\Phi(r)$, where $N$ is the standard Gaussian pdf, $\Phi$ is the standard Gaussian cdf. We define $\alpha(r) = \Phi(r) + rh(r) - h(r)^2$.

As $\sigma(x)^2 = \sigma(-x)^2$, it suffices to show $\alpha(r(x)) \geq \alpha(r(-x))$ for $x > 0$. To show this, we first show that $r(x) \geq r(-x)$ for $x > 0$, then show that $\alpha(r)$ is monotonically increasing.
$$r(x) = \mu(x)/\sigma(x) = \mu(-x)/\sigma(-x) + 2\mu_u x/\sigma(-x) \geq \mu(-x)/\sigma(-x) = r(-x).$$
The inequality uses that both $\mu_u$ and $x$ are non-negative. It remains to show that $\alpha(r)$ is monotonically increasing. A straightforward calculation shows that,
$$\alpha'(r) = 2h(r)(1 - \Phi(r)).$$
As $1 - \Phi(r) > 0$, we must show $h(r) \geq 0$. We have $\lim_{r \to -\infty} h(r) = 0$ and $h'(r) = \Phi(r) > 0$, implying $h(r) > 0$. We conclude $\alpha'(r) > 0$ for all $r$, showing that $v_{\mu_u}(x) \geq v_{\mu_u}(-x)$ for $x \geq 0$.

To complete the proof, we must show that $v(x)$ is monotonically increasing for $x \geq 0$. As $\sigma(x)^2$ is increasing as a function of $x$ and $\alpha(r)$ is increasing as a function of $r$, $v(x)$ is increasing as a function of $x$ whenever $r(x)$ is increasing as a function of $x$. As $r'(x) = \frac{\sigma_v^2 \mu_u - \sigma_u^2 \mu_v x}{\sigma(x)^3}$, this completes the proof if $\sigma_v^2 \mu_u - \sigma_u^2 \mu_v x \geq 0$. In particular, we need only consider cases when $\mu_v > 0$. In this case, we write,
$$v(x) = \mu(x)^2 \beta(r(x)) \tag{6}$$
where $\beta(r) = \alpha(r)/r^2$. Also in this region, we have the inequality,
$$r'(x)\sigma(x) = \frac{\sigma_v^2 \mu_u - \sigma_u^2 \mu_v x}{\sigma_u^2 x^2 + \sigma_v^2} \leq \frac{\sigma_v^2 \mu_u}{\sigma_u^2 x^2 + \sigma_v^2} \leq \frac{\sigma_v^2 \mu_u}{\sigma_v^2} = \mu_u,$$
which leads to $r'(x) \leq \mu_u/\sigma(x)$.

Differentiating equation 6,
$$v'(x) = 2\mu_u \mu(x) \beta(r(x)) + \mu(x)^2 \left( \frac{\sigma_v^2 \mu_u - \sigma_u^2 \mu_v x}{\sigma(x)^3} \right) \beta'(r(x))$$
$$\geq 2\mu_u \mu(x) \left( \beta(r(x)) + \frac{1}{2} r(x) \beta'(r(x)) \right).$$
The inequality uses that $r(x) > 0$, so that by lemma 4, $\beta'(r(x)) < 0$. It suffices to show that $\beta(r) + \frac{1}{2} r \beta'(r) > 0$ for $r > 0$.
$$\beta(r) + \frac{1}{2} r \beta'(r) = \beta(r) + \frac{1}{2} r \frac{d}{dr} \left( \frac{\alpha(r)}{r^2} \right) = \frac{\alpha(r)}{r^2} + \frac{1}{2} r \frac{\alpha'(r) r^2 - 2r\alpha(r)}{r^4} = \frac{\alpha'(r)}{2r} \geq 0.$$
We conclude that $v'(x) \geq 0$ for $x \geq 0$, implying that $v(x)$ is monotonically increasing in this region. This completes the proof that $v_{\mu_u}(x) \in \mathcal{T}_1$ for $\mu_u > 0$.

**Lemma 4.** *For $\beta$ defined as in the proof of lemma 2 and for $r > 0$, $\beta'(r) < 0$*

*Proof.* For $r \neq 0$, $\beta'(r) = \left( -2\Phi(r) + 2N(r)^2 + 2N(r)\Phi(r) \right)/r^3$. As $r > 0$,
$$\beta'(r) \leq 0 \Leftrightarrow I(r) := -\Phi(r) + N(r)^2 + N(r)r\Phi(r) \leq 0.$$
Rearranging [1, 7.1.13] yields:
$$1 - \frac{2}{r + \sqrt{r^2 + 8/\pi}} N(r) \leq \Phi(r) < 1 - \frac{2}{r + \sqrt{r^2 + 4}} N(r). \tag{7}$$

for $r \geq 0$.

$$I(r) = -\Phi(r) + N(r)^2 + rN(r)\Phi(r)$$

$$\leq -\Phi(r) + N(r)^2 + rN(r)\left(1 - \frac{2}{r + \sqrt{r^2 + 4}}N(r)\right)$$

$$\leq -1 + \frac{2}{r + \sqrt{r^2 + 8/\pi}}N(r) + N(r)^2 + rN(r)\left(1 - \frac{2}{r + \sqrt{r^2 + 4}}N(r)\right)$$

$$= -1 + \frac{2}{r + \sqrt{r^2 + 8/\pi}}N(r) + rN(r) + N(r)^2\left(1 - \frac{2r}{r + \sqrt{r^2 + 4}}\right) \qquad (8)$$

We now make use of numerous crude bounds which hold for $r > 0$:

1. $N(r) \leq 1/\sqrt{2\pi}$,

2. $\frac{2}{r + \sqrt{r^2 + 8/\pi}} \leq \sqrt{\pi/2}$,

3. $rN(r) \leq 1/\sqrt{2\pi e}$

4. $\frac{2r}{r + \sqrt{r^2 + 4}} \geq 0$.

Plugging these into equation 8,

$$I(r) \leq -1 + \frac{\sqrt{\pi/2}}{\sqrt{2\pi}} + \frac{1}{\sqrt{2\pi e}} + \frac{1}{2\pi} = -\frac{1}{2} + \frac{1}{\sqrt{2\pi e}} + \frac{1}{2\pi} \approx -0.098 < 0. \qquad \square$$

### C.3 Proof of Lemma 3

*Proof.* Recall that

$$\mathcal{T}_1 = \{f \geq 0 : \forall 0 \leq b < a, f(a) \geq f(-a) \text{ and } f(b) \leq f(a)\}$$

and

$$\mathcal{T}_2 = \{f \geq 0 : \forall a < b \leq 0, f(a) \geq f(-a) \text{ and } f(b) \leq f(a)\}.$$

First, note that $\mathcal{T}_1, \mathcal{T}_2$ and the set of non-negative convex functions, $\mathcal{C}$ are all closed under addition and positive scalar multiplication. We can therefore write $f$ as a sum of three functions, $f(x) = t_1(x) + t_2(x) + s(x)$ with $t_1 \in \mathcal{T}_1, t_2 \in \mathcal{T}_2$ and $s \in \mathcal{C}$. We prove the case when $a \leq c \leq 0 \leq -c \leq b$. The case $a \leq -c \leq 0 \leq c \leq b$ follows a symmetric argument.

$$\begin{aligned}
f(c) &= t_1(c) + t_2(c) + s(c) \quad \text{(def.)} \\
&\leq t_1(c) + t_2(a) + s(c) \quad \text{(second condition for } \mathcal{T}_2\text{)} \\
&\leq t_1(-c) + t_2(a) + s(c) \quad \text{(first condition for } \mathcal{T}_1\text{)} \\
&\leq t_1(b) + t_2(a) + s(c) \quad \text{(second condition for } \mathcal{T}_1\text{)} \\
&\leq t_1(b) + t_2(a) + \max(s(a), s(b)) \quad \text{(}s \text{ convex)} \\
&\leq t_1(b) + t_2(a) + s(a) + s(b) \\
&\leq t_1(a) + t_1(b) + t_2(a) + t_2(b) + s(a) + s(b) \quad \text{(non-negativity)} \\
&= f(a) + f(b). \qquad \square
\end{aligned}$$

## D  Proof of Theorem 3

We now restate and prove Theorem 3 from the main body:

**Theorem 3.** *Let $A \subset \mathbb{R}^D$ be compact, and let $C(A)$ be the space of continuous functions on $A$ to $\mathbb{R}$. Similarly, let $C^+(A)$ be the space of continuous functions on $A$ to $\mathbb{R}_{\geq 0}$. Then for any $g \in C(A)$ and $h \in C^+(A)$, and any $\epsilon > 0$, for both the mean-field Gaussian and MC dropout families, there exists a 2-hidden layer ReLU NN such that*

$$\sup_{\mathbf{x} \in A} |\mathbb{E}[f(\mathbf{x})] - g(\mathbf{x})| < \epsilon \quad \text{and} \quad \sup_{\mathbf{x} \in A} |\mathbb{V}[f(\mathbf{x})] - h(\mathbf{x})| < \epsilon,$$

*where $f(\mathbf{x})$ is the (stochastic) output of the network.*

Our proof will make use of the standard universal approximation theorem for deterministic NNs as given in Leshno et al. [8]:

**Theorem 4** (Universal approximation for deterministic NNs)**.** *Let* $\psi(a) = \max(0, a)$*. Then for every* $g \in C(\mathbb{R}^D)$ *and every compact set* $A \subset \mathbb{R}^D$*, for any* $\epsilon > 0$ *there exists a function* $f \in S$ *such that* $\|g - f\|_\infty < \epsilon$*. Here*

$$S = \left\{ \sum_{i=1}^{I} w_i \psi \left( \sum_{d=1}^{D} u_{i,d} x_d + v_i \right) : I \in \mathbb{N}, w_i, u_{i,d}, v_i \in \mathbb{R} \right\}.$$

We first prove a useful lemma.

**Lemma 5.** *Let* $\psi(a) = \max(0, a)$*. Let* $a$ *be a random variable with finite first two moments. Then* $\mathbb{V}[\psi(a)] \leq \mathbb{V}[a]$*.*

*Proof.* For all $x, y \in \mathbb{R}$, we have $|x - y|^2 \geq |\psi(x) - \psi(y)|^2$. Consider two i.i.d. copies of any random variable with finite first two moments, denoted $a_1$ and $a_2$. Then

$$\mathbb{V}[a_1] = \mathbb{E}\left[a_1^2\right] - \mathbb{E}[a_1]^2 = \frac{1}{2}\mathbb{E}\left[a_1^2 + a_2^2 - 2a_1 a_2\right] = \frac{1}{2}\mathbb{E}\left[|a_1 - a_2|^2\right] \geq \frac{1}{2}\mathbb{E}\left[|\psi(a_1) - \psi(a_2)|^2\right]$$
$$= \mathbb{V}[\psi(a_1)]. \qquad \square$$

## D.1 Proof of Theorem 3 for $\mathcal{Q}_{\text{FFG}}$

We prove theorem 3 for the fully-factorised Gaussian approximating family. We begin by proving results about 1HL networks within this family. The overall goal of these results is lemma 8, which informally says that for any set of mean parameters for the weights, we can find a setting of the standard deviations of the weights, such that the mean output of the network is close to the output of the deterministic network, with weights equal to the mean parameters. Our proof of this proceeds in 3 parts: First, in lemma 9, we show that by making the standard deviation parameters sufficiently small, we can ensure that the variance of the output of the network is uniformly small on some compact set $A$. Next, in lemma 7, we show that again by choosing the standard deviation sufficiently small, we can show that most of the sample functions of the 1HL network are close to the function that would be obtained by using the mean parameters. Finally, in the proof of lemma 8, we use Chebyshev's inequality and the triangle inequality to conclude that the mean of the network must also be close to the function defined by the mean parameters.

These networks will be used to construct the desired 2HL network.

**Notation** Consider a 1HL ReLU NN with input $\mathbf{x} \in \mathbb{R}^D$ and output $\mathbf{f} \in \mathbb{R}^K$. Let the network have $I$ hidden units and be parameterised by input weights $U \in \mathbb{R}^{I \times D}$, input biases $v \in \mathbb{R}^I$, output weights $W \in \mathbb{R}^{K \times I}$ and output biases $b \in \mathbb{R}^K$. Let $\theta = (U, v, W, b)$. Denote the $k^{\text{th}}$ output of the network by $f_\theta^{(k)}(\mathbf{x})$. Consider a factorised Gaussian distribution over the parameters $\theta$ in the network. Let the means of the Gaussians be denoted $\boldsymbol{\mu} = (\mu_U, \mu_v, \mu_W, \mu_b)$, where e.g. $\mu_U$ is a matrix whose elements are the means of $U$. Each mean is always taken to be $\in \mathbb{R}$. Let the standard deviations be denoted $\boldsymbol{\sigma} = (\sigma_U, \sigma_v, \sigma_W, \sigma_b)$. Each standard deviation is always taken to be $\in \mathbb{R}_{>0}$.

The following lemma states that we can make the output of a 1HL BNN have low variance by setting the standard deviation of the weights to be small.

**Lemma 6.** *Let* $A \subset \mathbb{R}^D$ *be a compact set and* $f_\theta^{(k)}(\mathbf{x})$ *be the* $k^{th}$ *output of a 1HL ReLU NN with a mean-field Gaussian distribution mapping from* $A \to \mathbb{R}$*. Fix any* $\boldsymbol{\mu}$ *and any* $\epsilon > 0$*. Let all the standard deviations in* $\boldsymbol{\sigma}$ *be equal to a shared constant* $\sigma > 0$*. Then there exists* $\sigma' > 0$ *such that for all* $\sigma < \sigma'$ *and for all* $\mathbf{x} \in A$*,* $\mathbb{V}[\psi(f_\theta^{(k)}(\mathbf{x}))] < \epsilon$ *for all* $1 \leq k \leq K$*.*

*Proof.* Define $a_i = \sum_{d=1}^{D} u_{i,d} x_d + v_i$, so that $f_\theta^{(k)}(\mathbf{x}) = \sum_{i=1}^{I} w_{k,i} \psi(a_i) + b_k$. Then

$$
\begin{aligned}
\mathbb{V}[f_\theta^{(k)}(\mathbf{x})] &= \mathbb{V}\left[\sum_{i=1}^{I} w_{k,i}\psi(a_i)\right] + \sigma^2 \\
&= \sum_{i=1}^{I}\sum_{j=1}^{I} \mathrm{Cov}\left(w_{k,i}\psi(a_i), w_{k,j}\psi(a_j)\right) + \sigma^2 \\
&\leq \sum_{i=1}^{I}\sum_{j=1}^{I} |\mathrm{Cov}\left(w_{k,i}\psi(a_i), w_{k,j}\psi(a_j)\right)| + \sigma^2 \\
&\leq \sum_{i=1}^{I}\sum_{j=1}^{I} \sqrt{\mathbb{V}[w_{k,i}\psi(a_i)]\mathbb{V}[w_{k,j}\psi(a_j)]} + \sigma^2,
\end{aligned}
$$

where the final line follows from the Cauchy–Schwarz inequality. We now analyse each of the constituent terms. Since $w_{k,i}$ and $\psi(a_i)$ are independent,

$$
\mathbb{V}[w_{k,i}\psi(a_i)] = \mu_{w_{k,i}}^2 \mathbb{V}[\psi(a_i)] + \mathbb{E}[\psi(a_i)]^2 \sigma^2 + \sigma^2 \mathbb{V}[\psi(a_i)].
$$

As $A$ is compact, it is bounded, so there exists an $M$ such that $|x_d| \leq M$ for all $1 \leq d \leq D$. Using lemma 5, and the mean-field assumptions,

$$
\mathbb{V}[\psi(a_i)] \leq \mathbb{V}[a_i] = \sigma^2 \left(\sum_{d=1}^{D} x_d^2 + 1\right) \leq \sigma^2(DM^2 + 1).
$$

Since $a_i$ is a linear combination of Gaussian random variables, we have that $a_i \sim \mathcal{N}(\mu_{a_i}, \sigma_{a_i}^2)$, where $\mu_{a_i} = \sum_{d=1}^{D} \mu_{u_{i,d}} x_d + \mu_{v_i}$ and $\sigma_{a_i}^2 = \sigma^2 \left(\sum_{d=1}^{D} x_d^2 + 1\right)$. Therefore, we have that [3]

$$
\begin{aligned}
\mathbb{E}[\psi(a_i)]^2 &= \left(\mu_{a_i}\Phi\left(\frac{\mu_{a_i}}{\sigma_{a_i}}\right) + \sigma_{a_i} N\left(\frac{\mu_{a_i}}{\sigma_{a_i}}\right)\right)^2 \leq \left(|\mu_{a_i}|\Phi\left(\frac{\mu_{a_i}}{\sigma_{a_i}}\right) + \sigma_{a_i} N\left(\frac{\mu_{a_i}}{\sigma_{a_i}}\right)\right)^2 \\
&\leq \left(|\mu_{a_i}| + \frac{\sigma_{a_i}}{\sqrt{2\pi}}\right)^2.
\end{aligned}
$$

We can then upper bound $\mathbb{V}[w_{k,i}\psi(a_i)]$ as follows:

$$
\begin{aligned}
\mathbb{V}[w_{k,i}\psi(a_i)] &\leq \mu_{w_{k,i}}^2 \sigma^2(DM^2 + 1) + \left(|\mu_{a_i}| + \frac{\sigma_{a_i}}{\sqrt{2\pi}}\right)^2 \sigma^2 + \sigma^4(DM^2 + 1) \\
&\leq \mu_{w_{k,i}}^2 \sigma^2(DM^2 + 1) + \left(M\sum_{d=1}^{D} |\mu_{u_{i,d}}| + |\mu_{v_i}| + \frac{\sqrt{\sigma^2(M^2 D + 1)}}{\sqrt{2\pi}}\right)^2 \sigma^2 + \sigma^4(DM^2 + 1) \\
&:= v_{k,i}(\sigma).
\end{aligned}
$$

The second inequality follows since $A$ is compact and we have $|\mu_{a_i}| \leq M\sum_{d=1}^{D} |\mu_{u_{i,d}}| + |\mu_{v_i}|$. Note that the upper bound $v_{k,i}(\sigma)$ is continuous and monotonically increasing in $\sigma$, and $v_{k,i}(0) = 0$. We can then upper bound the variance of the output:

$$
\mathbb{V}[f_\theta^{(k)}(\mathbf{x})] \leq \sum_{i=1}^{I}\sum_{j=1}^{I} \sqrt{v_{k,i}(\sigma)v_{k,j}(\sigma)} + \sigma^2.
$$

We then choose $\sigma'$ such that for all $1 \leq k \leq K$ and for all $1 \leq i \leq I$, $v_{k,i}(\sigma') < \frac{\epsilon}{2I^2}$, and such that $\sigma'^2 < \frac{\epsilon}{2}$. Then

$$
\mathbb{V}[f_\theta^{(k)}(\mathbf{x})] \leq I^2 \frac{\epsilon}{2I^2} + \sigma'^2 < \epsilon
$$

for $1 \leq k \leq K$. Finally, applying lemma 5, we have $\mathbb{V}[\psi(f_\theta^{(k)}(\mathbf{x}))] < \epsilon$ for $1 \leq k \leq K$. $\qquad\square$

The following lemma states that by setting the standard deviation of the weights to be sufficiently small, we can with high probability make the sampled BNN output close to the BNN output evaluated at the mean parameters.

**Lemma 7.** *Let $A \subset \mathbb{R}^D$ be any compact set. Fix any $\boldsymbol{\mu}$ and any $\epsilon, \delta > 0$. Let all the standard deviations in $\boldsymbol{\sigma}$ be equal to a shared constant $\sigma > 0$. Then there exists $\sigma' > 0$ such that for all $\sigma < \sigma'$, and for any $\mathbf{x} \in A$,*

$$\Pr\left(|\psi(f_{\boldsymbol{\mu}}^{(k)}(\mathbf{x})) - \psi(f_{\theta}^{(k)}(\mathbf{x}))| > \epsilon\right) < \delta$$

*for all $1 \leq k \leq K$.*

*Proof.* Let $\theta \in \mathbb{R}^P$. We first note that $\psi(f_{\theta}^{(k)}(\mathbf{x}))$ is continuous as a function from $A \times \mathbb{R}^P \to \mathbb{R}$, under the metric topology induced by the Euclidean metric on $A \times \mathbb{R}^P$. Next, define a ball in parameter space

$$B_\gamma = \{\theta : \|\theta - \boldsymbol{\mu}\|_2 < \gamma\}.$$

Consider the closed ball of unit radius around $\boldsymbol{\mu}$, $\bar{B}_1$. Note that $\bar{B}_1$ is compact, and therefore $A \times \bar{B}_1$ is compact as a product of compact spaces.

Since a continuous map from a compact metric space to another metric space is uniformly continuous, given $\epsilon > 0$, there exists a $0 < \tau < 1$ such that for all pairs $(\mathbf{x}_1, \theta_1), (\mathbf{x}_2, \theta_2) \in A \times \bar{B}_1$ such that $d((\mathbf{x}_1, \theta_1), (\mathbf{x}_2, \theta_2)) < \tau$, $|\psi(f_{\theta_1}^{(k)}(\mathbf{x}_1)) - \psi(f_{\theta_2}^{(k)}(\mathbf{x}_2))| < \epsilon$. Here $d(\cdot, \cdot)$ is the Euclidean metric on $A \times \mathbb{R}^P$. Since this is true for all $1 \leq k \leq K$, we can find a $0 < \tau < 1$ such that $|\psi(f_{\theta_1}^{(k)}(\mathbf{x}_1)) - \psi(f_{\theta_2}^{(k)}(\mathbf{x}_2))| < \epsilon$ holds for all $k$ simultaneously, by taking the minimum of the $\tau$ over $k$.

Now choose $\sigma' > 0$ such that for all $\sigma < \sigma'$, $\Pr(\theta \in B_\tau) > 1 - \delta$. This event implies $d((\mathbf{x}, \theta), (\mathbf{x}, \boldsymbol{\mu})) = \|\theta - \boldsymbol{\mu}\|_2 < \tau$. Furthermore, $\theta \in \bar{B}_1$, since $\tau < 1$. Hence $|\psi(f_{\boldsymbol{\mu}}^{(k)}(\mathbf{x})) - \psi(f_{\theta}^{(k)}(\mathbf{x}))| < \epsilon$ holds for all $1 \leq k \leq K$.

$\square$

The following lemma shows that for 1HL networks, we can make $\mathbb{E}\left[\psi(f_{\theta}^{(k)})\right]$ (the mean BNN output) close to $\psi(f_{\boldsymbol{\mu}}^{(k)})$ (the BNN output evaluated at the mean parameter settings) by choosing the standard deviation of the weights to be sufficiently small.

**Lemma 8.** *Let $A \subset \mathbb{R}^D$ be any compact set. Then, for any $\epsilon > 0$ and any $\boldsymbol{\mu}$, there exists a $\sigma_1 > 0$ such that for any shared standard deviation $\sigma < \sigma_1$,*

$$\left\|\mathbb{E}\left[\psi(f_{\theta}^{(k)})\right] - \psi(f_{\boldsymbol{\mu}}^{(k)})\right\|_\infty < \epsilon$$

*for all $1 \leq k \leq K$.*

*Proof.* For all $\mathbf{x} \in A$ and any $\theta^*$, by the triangle inequality

$$\left|\mathbb{E}\left[\psi(f_{\theta}^{(k)}(\mathbf{x}))\right] - \psi(f_{\boldsymbol{\mu}}^{(k)}(\mathbf{x}))\right| \leq \left|\mathbb{E}\left[\psi(f_{\theta}^{(k)}(\mathbf{x}))\right] - \psi(f_{\theta^*}^{(k)}(\mathbf{x}))\right| + \left|\psi(f_{\boldsymbol{\mu}}^{(k)}(\mathbf{x})) - \psi(f_{\theta^*}^{(k)}(\mathbf{x}))\right|.$$

Applying lemma 7 with $\epsilon' = \epsilon/2$ and $\delta = 1/4$, we can find a $\sigma'$ such that for all $\sigma < \sigma'$, $\left|\psi(f_{\boldsymbol{\mu}}^{(k)}(\mathbf{x})) - \psi(f_{\theta}^{(k)}(\mathbf{x}))\right| \leq \epsilon/2$ with probability at least $3/4$. By lemma 6, we can find a $\sigma''$ such that for all $\sigma < \sigma''$, $\mathbb{V}[\psi(f_{\theta}^{(k)}(\mathbf{x}))] < \frac{\epsilon^2}{16K}$. Choose $0 < \sigma < \min(\sigma', \sigma'')$. We can apply Chebyshev's inequality to each random variable $\psi(f_{\theta}^{(k)}(\mathbf{x}))$,

$$\Pr\left[\left|\psi(f_{\theta}^{(k)}(\mathbf{x})) - \mathbb{E}\left[\psi(f_{\theta}^{(k)}(\mathbf{x}))\right]\right| > \epsilon/2\right] < \frac{1}{4K}.$$

Applying the union bound, the probability that $|\psi(f_{\theta}^{(k)}(\mathbf{x})) - \mathbb{E}\left[\psi(f_{\theta}^{(k)}(\mathbf{x}))\right]| \leq \epsilon/2$ for all $k$ simultaneously is at least $3/4$. Therefore, for any $\mathbf{x}$ we can find a $\theta^*$ such that $|\psi(f_{\theta^*}^{(k)}(\mathbf{x})) -$

$\mathbb{E}\left[\psi(f_{\theta}^{(k)}(\mathbf{x}))\right] | \leq \epsilon/2$ and $\left|\psi(f_{\boldsymbol{\mu}}^{(k)}(\mathbf{x})) - \psi(f_{\theta^*}^{(k)}(\mathbf{x}))\right| \leq \epsilon/2$ simultaneously because both events occur with probability at least $1/2$ and therefore have a non-empty intersection. Therefore for all $\mathbf{x}$ and all $k$

$$\left|\mathbb{E}\left[\psi(f_{\theta}^{(k)}(\mathbf{x}))\right] - \psi(f_{\boldsymbol{\mu}}^{(k)}(\mathbf{x}))\right| \leq \epsilon. \qquad \square$$

We can now complete the proof of theorem 3 for $\mathcal{Q}_{\text{FFG}}$.

*Proof of theorem 3.* Consider the case of a 2-hidden layer ReLU Bayesian neural network with 2 units in the second hidden layer. Denote the inputs to these units as $f_{\theta}^{(1)}(\mathbf{x})$ and $f_{\theta}^{(2)}(\mathbf{x})$ respectively, where $\theta$ are the parameters in the bottom two weight matrices and biases of the network. The output of the network can then be written as,

$$f(\mathbf{x}) = s_1\psi(f_{\theta}^{(1)}(\mathbf{x})) + s_2\psi(f_{\theta}^{(2)}(\mathbf{x})) + t \qquad (9)$$

where the $s_i$ are the weights in the final layer and $t$ is the bias. Taking expectations on both sides,

$$\mathbb{E}[f(\mathbf{x})] = \mathbb{E}\left[s_1\psi(f_{\theta}^{(1)}(\mathbf{x}))\right] + \mathbb{E}\left[s_2\psi(f_{\theta}^{(2)}(\mathbf{x}))\right] + \mathbb{E}[t]$$

Choose $\mu_{s_1} = 1, \mu_{s_2} = 0$, and note that $s_1$ is independent of $\theta$ by the mean field assumption. Then

$$\mathbb{E}[f(\mathbf{x})] = \mathbb{E}\left[\psi(f_{\theta}^{(1)}(\mathbf{x}))\right] + \mathbb{E}[t]. \qquad (10)$$

Define $\mu_t = -\min_{\mathbf{x}' \in A} g(\mathbf{x}')$ (as $A$ is compact and $g$ is continuous, this minimum is well-defined). Define $\tilde{g}(\mathbf{x}) \geq 0$ to be $g(\mathbf{x}) - \min_{\mathbf{x}' \in A} g(\mathbf{x}')$. By the universal approximation theorem (theorem 4) we can find a setting of the mean parameters, $\boldsymbol{\mu}$ in the first two layers (i.e. excluding the parameters of the distributions on $s_1, s_2$ and $t$) such that

$$\|f_{\boldsymbol{\mu}}^{(1)} - \tilde{g}\|_{\infty} < \epsilon/2 \quad \text{and} \quad \|f_{\boldsymbol{\mu}}^{(2)} - \sqrt{h}\|_{\infty} < \epsilon/2.$$

This can be done by splitting the neurons in the first hidden layer into two sets, where the first and second set are responsible for $f^{(1)}, f^{(2)}$ respectively, and the weights from each set to the output of the other set are zero. Since $\tilde{g}(\mathbf{x}) > 0$, applying the ReLU can only make $f^{(1)}$ closer to $\tilde{g}$. Hence $\|\psi(f_{\boldsymbol{\mu}}^{(1)}) - \tilde{g}\|_{\infty} < \epsilon/2$.

By lemma 8, we can find a $\sigma_1 > 0$ for this $\boldsymbol{\mu}$ such that when the standard deviations in the first two layers are set to any shared constant $\sigma < \sigma_1$,

$$\left\|\mathbb{E}\left[\psi(f_{\theta}^{(1)})\right] - \psi(f_{\boldsymbol{\mu}}^{(1)})\right\|_{\infty} < \epsilon/2.$$

By the triangle inequality, $\left\|\mathbb{E}\left[\psi(f_{\theta}^{(1)})\right] - \tilde{g}\right\|_{\infty} < \epsilon$. Combining with equation 10, it follows that the expectation can approximate any continuous function $g$.

We now consider the variance of equation 9.

$$\mathbb{V}[f(\mathbf{x})] = \mathbb{V}[s_1\psi(f_{\theta}^{(1)}(\mathbf{x})) + s_2\psi(f_{\theta}^{(2)}(\mathbf{x}))] + \mathbb{V}[t]$$
$$= \mathbb{V}[s_1\psi(f_{\theta}^{(1)}(\mathbf{x}))] + \mathbb{V}[s_2\psi(f_{\theta}^{(2)}(\mathbf{x}))] + 2\text{Cov}(s_1\psi(f_{\theta}^{(1)}(\mathbf{x})), s_2\psi(f_{\theta}^{(2)}(\mathbf{x}))) + \sigma_t^2.$$

Choose $\sigma_t^2 = \epsilon$. We now consider $\mathbb{V}[s_1\psi(f_{\theta}^{(1)}(\mathbf{x}))]$. As $s_1$ is independent of $\theta$,

$$\mathbb{V}[s_1\psi(f_{\theta}^{(1)}(\mathbf{x}))] = \mu_{s_1}^2 \mathbb{V}[\psi(f_{\theta}^{(1)}(\mathbf{x}))] + \sigma_{s_1}^2 \mathbb{E}\left[\psi(f_{\theta}^{(1)}(\mathbf{x}))\right]^2 + \mathbb{V}[\psi(f_{\theta}^{(1)}(\mathbf{x}))]\sigma_{s_1}^2.$$

Recall $\mu_{s_1} = 1$ and choose $\sigma_{s_1}^2 = \min\left(1, \epsilon/\left(\max_{x \in A} \mathbb{E}\left[\psi(f_{\theta}^{(1)}(\mathbf{x}))\right]^2\right)\right)$, then

$$\mathbb{V}[s_1\psi(f_{\theta}^{(1)}(\mathbf{x}))] \leq 2\mathbb{V}[\psi(f_{\theta}^{(1)}(\mathbf{x}))] + \epsilon.$$

By lemma 6, we can find a $\sigma_2$ such that for any $\sigma < \sigma_2$, $\mathbb{V}[\psi(f_{\theta}^{(1)}(\mathbf{x}))] \leq \epsilon$. For any such $\sigma$, $\mathbb{V}[s_1\psi(f_{\theta}^{(1)}(\mathbf{x}))] \leq 3\epsilon$.

We now choose $\sigma_{s_2}^2 = 1$ and consider

$$\mathbb{V}[s_2\psi(f_\theta^{(2)}(\mathbf{x}))] = \mu_{s_2}^2 \mathbb{V}[\psi(f_\theta^{(2)}(\mathbf{x}))] + \sigma_{s_2}^2 \mathbb{E}\left[\psi(f_\theta^{(2)}(\mathbf{x}))\right]^2 + \sigma_{s_2}^2 \mathbb{V}[\psi(f_\theta^{(2)}(\mathbf{x}))]$$

$$= \mathbb{E}\left[\psi(f_\theta^{(2)}(\mathbf{x}))\right]^2 + \mathbb{V}[\psi(f_\theta^{(2)}(\mathbf{x}))].$$

By lemma 6, we can find a $\sigma_3$ such that for any $\sigma < \sigma_3$, $\mathbb{V}[\psi(f_\theta^{(2)}(\mathbf{x}))] < \epsilon$.

By the universal function approximator theorem (theorem 4) we can find a setting of the mean parameters, $\boldsymbol{\mu}$ in the first two layers such that $\|f_{\boldsymbol{\mu}}^{(2)} - \sqrt{h}\|_\infty < \epsilon/2$. Since $\sqrt{h(\mathbf{x})} > 0$, the ReLU can only make $f^{(2)}$ closer to $\sqrt{h}$, $\|\psi(f_{\boldsymbol{\mu}}^{(2)}) - \sqrt{h})\|_\infty < \epsilon/2$.

By lemma 8, we can find a setting of $\sigma$ for this $\boldsymbol{\mu}$ such that

$$\left\|\mathbb{E}\left[\psi(f_\theta^{(2)})\right] - \psi(f_{\boldsymbol{\mu}}^{(2)})\right\|_\infty < \epsilon/2.$$

By the triangle inequality,

$$\left\|\mathbb{E}\left[\psi(f_\theta^{(2)})\right] - \sqrt{h}\right\|_\infty < \epsilon.$$

This implies,

$$\left\|\mathbb{E}\left[\psi(f_\theta^{(2)})\right]^2 - h\right\|_\infty = \left\|\left(\mathbb{E}\left[\psi(f_\theta^{(2)})\right] - \sqrt{h}\right)\left(\mathbb{E}\left[\psi(f_\theta^{(2)})\right] + \sqrt{h}\right)\right\|_\infty$$

$$\leq \epsilon \left\|\mathbb{E}\left[\psi(f_\theta^{(2)})\right] + \sqrt{h}\right\|_\infty$$

$$\leq \epsilon(2\|\sqrt{h}\|_\infty + \epsilon)$$

We therefore have,

$$\|\mathbb{V}[f] - h\|_\infty \leq E(\epsilon) + 2\mathrm{Cov}(s_1\psi(f_\theta^{(1)}(\mathbf{x})), s_2\psi(f_\theta^{(2)}(\mathbf{x})))$$

$$\leq E(\epsilon) + 2\sqrt{\mathbb{V}[s_1\psi(f_\theta^{(1)}(\mathbf{x}))]\mathbb{V}[s_2\psi(f_\theta^{(2)}(\mathbf{x}))]}$$

$$\leq E(\epsilon) + C\sqrt{\epsilon}$$

where the first inequality is Cauchy-Schwarz, and $E(\epsilon)$ is a function that tends to zero with $\epsilon$ and $C$ is a constant. The theorem follows by choosing $\sigma < \min\{\sigma_1, \sigma_2, \sigma_3\}$. $\qquad\square$

The construction in our proof used a 2HL BNN with only two neurons in the second hidden layer. The construction still works for wider hidden layers, by setting the unused neurons to have zero mean and sufficiently small variance.

An analogous statement to theorem 3 for networks with more than two hidden layers can be proved inductively: applying theorem 3 for 2HL BNNs we can choose the variance to be uniformly small, thus satisfying the condition stated in lemma 6. The proof of lemma 7 applies equally for the output of 2HL BNNs. The rest of the proof then follows as stated.

### D.2 Proof of theorem 3 for MCDO

In order to prove the universality result for deep dropout, we first prove two lemmas about 1HL dropout networks. The following lemma states that the mean of a 1HL dropout network is a universal function approximator, while its variance can simultaneously be made arbitrarily small.

**Lemma 9.** *Consider any $\epsilon > 0$ and any continuous function, $m$ mapping from a compact subset, $A$ of $\mathbb{R}^D$ to $\mathbb{R}$. Then there exists a (random) ReLU neural network of the form*

$$f(x) = \sum_{i=1}^{I} w_i\gamma_i\psi\left(\sum_{d=1}^{D} u_{i,d}x_d + v_i\right) + b$$

*with $\gamma_i \overset{\text{i.i.d.}}{\sim} \mathrm{Bern}(1-p)$ such that $\|\mathbb{E}[f] - m\|_\infty < \epsilon$ and $\|\mathbb{V}[f]\|_\infty \leq \epsilon$.*

*Proof.* By the universal approximation theorem Leshno et al. [8], there exists a $J$ and 1HL network of the form,

$$g(x) = \sum_{j=1}^{J} \tilde{w}_j \psi \left( \sum_{d=1}^{D} \tilde{u}_{j,d} x_d + v_j \right) + b,$$

such that $\|g - m\|_\infty \leq \epsilon$. Define the dropout network,

$$f^{(1)}(x) = \sum_{j=1}^{J} \frac{\tilde{w}_j}{1-p} \psi \left( \sum_{d=1}^{D} \tilde{u}_{j,d} x_d + v_j \right) + b.$$

Then $\mathbb{E}\left[f^{(1)}\right] = g$, so that $\|\mathbb{E}\left[f^{(1)}\right] - m\|_\infty \leq \epsilon$. Let $S = \|\mathbb{V}[f^{(1)}]\|_\infty < \infty$.

Define $f = \frac{1}{L} \sum_{\ell=1}^{L} f^{(1,\ell)}$ where each $f^{(1,\ell)}$ is an independent realisation of $f^{(1)}$. Then $\mathbb{E}[f] = g$ and $\mathbb{V}[f] = \frac{\mathbb{V}[f^{(1)}]}{\sqrt{L}} \leq \frac{S}{\sqrt{L}}$. $f$ can be realised by a dropout network by combining $L$ copies of $f^{(1)}$ together with identical weights within each copy and $0$ weights connecting the various copies. Choosing $L = (S/\epsilon)^2$ completes the proof. □

The following lemma states that the mean of the MCDO network can approximate any continuous *positive* function, after application of the ReLU non-linearity.

**Lemma 10.** *Given a positive mean function $m$ with $0 < \delta \leq \|m\|_\infty \leq \Delta$ and a stochastic process $f$ such that $\|\mathbb{E}[f] - m\|_\infty \leq \epsilon \leq \delta$ and $\|\mathbb{V}[f]\|_\infty \leq \epsilon$,*

$$\|\mathbb{E}[\psi(f)] - m\|_\infty \leq \epsilon + \frac{\sqrt{\epsilon^2 + \epsilon(\Delta+\epsilon)^2}}{\delta - \epsilon} = \mathcal{O}(\Delta\sqrt{\epsilon}/(\delta - \epsilon))$$

*and $\|\mathbb{V}[\psi(f)]\|_\infty \leq \epsilon$. In the big-O notation, we assume $\Delta$ is bounded below by a constant and $\epsilon, \delta$ are bounded above by a constant.*

*Proof.* The bound $\|\mathbb{V}[\psi(f)]\|_\infty \leq \epsilon$ follows from lemma 5. We consider the expectation of $\psi(f(\mathbf{x}))$ for some arbitrary fixed $\mathbf{x}$,

$$\begin{aligned}
|\mathbb{E}[\psi(f(\mathbf{x}))] - m(\mathbf{x})| &= |\mathbb{E}[f(\mathbf{x})] - m(\mathbf{x}) - \mathbb{E}[\min(0, f(\mathbf{x}))]| \\
&\leq |\mathbb{E}[f(\mathbf{x})] - m(\mathbf{x})| + |\mathbb{E}[\min(0, f(\mathbf{x}))]| \\
&\leq \epsilon + |\mathbb{E}[\min(0, f(\mathbf{x}))]|.
\end{aligned}$$

We therefore bound $|\mathbb{E}[\min(0, f(\mathbf{x}))]|$.

$$|\mathbb{E}[\min(0, f(\mathbf{x}))]| = |\mathbb{E}[f(\mathbf{x})\mathbf{1}\{\mathbf{x} : f(\mathbf{x}) < 0\}]| \leq \sqrt{\mathbb{E}[f(\mathbf{x})^2]\Pr(f(\mathbf{x}) < 0)}.$$

The inequality uses Cauchy-Schwarz, that the square of an indicator function is itself and reinterprets the expectation of an indicator function as a probability. We bound the two terms on the RHS separately.

$$\mathbb{E}\left[f(\mathbf{x})^2\right] = \mathbb{V}[f(\mathbf{x})] + \mathbb{E}[f(\mathbf{x})]^2 \leq \epsilon + \mathbb{E}[f(\mathbf{x})]^2 \leq \epsilon + (m(\mathbf{x}) + \epsilon)^2 \leq \epsilon + (\Delta + \epsilon)^2$$

We use Chebyshev's inequality to bound the probability $f(x) < 0$,

$$\begin{aligned}
\Pr(f(\mathbf{x}) < 0) &\leq \Pr\left(|f(\mathbf{x}) - \mathbb{E}[f(\mathbf{x})]| > m(\mathbf{x}) - \epsilon\right) \\
&\leq \frac{\mathbb{V}[f(\mathbf{x})]}{(m(\mathbf{x}) - \epsilon)^2} \\
&\leq \frac{\epsilon}{(m(\mathbf{x}) - \epsilon)^2} \\
&\leq \frac{\epsilon}{(\delta - \epsilon)^2}.
\end{aligned}$$ □

Having collected the necessary lemmas, we provide a construction that proves theorem 3.

*Proof of theorem 3.* Consider a 2HL dropout NN. Let the pre-activations in the first hidden layer be collectively denoted $\mathbf{a}_1$, and the random dropout masks by $\boldsymbol{\epsilon}_1$. Let the second hidden layer have $I + 2$ hidden units. Let $\odot$ denote the elementwise product of two vectors of the same length. Define the pre-activations of two of the second hidden layer units by $a_v = \mathbf{w}_v^\top(\boldsymbol{\epsilon_1} \odot \psi(\mathbf{a}_1))$, i.e. both these hidden units have identical weight vectors $\mathbf{w}_v$ and dropout masks, and are hence the same random variable. Similarly, let the remaining $I$ second hidden layer pre-activations be defined by $a_m = \mathbf{w}_m^\top(\boldsymbol{\epsilon_1} \odot \psi(\mathbf{a}_1))$, again all being the same random variable. Furthermore, let $(\mathbf{w}_v)_i = 0$ whenever $(\mathbf{w}_m)_i \neq 0$ and vice versa, so that the first hidden layer neurons that influence $a_v$ and those that influence $a_m$ form disjoint sets. Then the output of the 2HL network is:

$$f = \epsilon_a w_{2,a}\psi(a_v) + \epsilon_b w_{2,b}\psi(a_v) + \sum_{i=1}^{I} \epsilon_i w_{2,i}\psi(a_m) + b_2,$$

where $\epsilon_a, \epsilon_b, \{\epsilon_i\}_{i=1}^{I}$ are the final layer dropout masks and $\{w_{2,i}\}_{i=1}^{I}, b_2$ are the final layer weights and bias.

We now make the choices $w_{2,a} = 1, w_{2,b} = -1, w_{2,i} = \alpha$, where $\alpha I = 1/(1-p)$. Then $\mathbb{E}[f] = \mathbb{E}[\psi(a_m)] + b_2$. Let $b_2 = \min_{\mathbf{x} \in A} g - \delta$, where $\delta > 0$ and the min exists due to compactness of $A$. Define $g' = g - b_2$. Since $a_m$ is just the output of a single-hidden layer dropout network, for any $\gamma' > 0$ we can use lemma 9 to choose $\|\mathbb{E}[a_m] - g'\|_\infty < \gamma'$ and $\|\mathbb{V}[a_m]\|_\infty < \gamma'$. Since $g'$ is bounded below by $\delta$ and bounded above by some $\Delta \in \mathbb{R}$ (by continuity of $g$ and compactness of $A$), we can then apply lemma 10 to obtain $\|\mathbb{E}[a_m] - g'\|_\infty = \mathcal{O}(\Delta\sqrt{\epsilon'}/(\delta - \epsilon'))$ and $\|\mathbb{V}[\psi(a_m)]\|_\infty < \gamma'$. We can use this to bound the error in the mean of the 2HL network output:

$$\|\mathbb{E}[f] - g\|_\infty = \|\mathbb{E}[\psi(a_m)] + b_2 - g\|_\infty = \|\mathbb{E}[\psi(a_m)] - g'\|_\infty = \mathcal{O}(\Delta\sqrt{\gamma'}/(\delta - \gamma')).$$

We can choose $\gamma'$ to depend on $\delta, \Delta$ such that $\|\mathbb{E}[f] - g\|_\infty < \gamma$, proving the first part of the theorem. Next, calculating the variance,

$$\mathbb{V}[f] = \mathbb{V}\left[(\epsilon_a - \epsilon_b)\psi(a_v) + \alpha\psi(a_m)\sum_{i=1}^{I}\epsilon_i\right] = \mathbb{V}[(\epsilon_a - \epsilon_b)\psi(a_v)] + \alpha^2\mathbb{V}\left[\psi(a_m)\sum_{i=1}^{I}\epsilon_i\right]. \tag{11}$$

Next we show that by taking $I$ sufficiently large, we can make the second term arbitrarily small. We have,

$$\mathbb{V}\left[\psi(a_m)\sum_{i=1}^{I}\epsilon_i\right] = \mathbb{V}[\psi(a_m)]\mathbb{V}\left[\sum_{i=1}^{I}\epsilon_i\right] + \mathbb{V}[\psi(a_m)]\mathbb{E}\left[\sum_{i=1}^{I}\epsilon_i\right]^2 + \mathbb{V}\left[\sum_{i=1}^{I}\epsilon_i\right]\mathbb{E}[\psi(a_m)]^2$$

$$= \mathbb{V}[\psi(a_m)]Ip(1-p) + \mathbb{V}[\psi(a_m)]I^2(1-p)^2 + Ip(1-p)\mathbb{E}[\psi(a_m)]^2$$

$$\leq \gamma' Ip(1-p) + \gamma' I^2(1-p)^2 + Ip(1-p)\mathbb{E}[\psi(a_m)]^2$$

The first two of these three terms can be made arbitrarily small by choosing $\gamma'$ sufficiently small. The third term, upon multiplying by $\alpha^2$, becomes

$$\alpha^2 Ip(1-p)\mathbb{E}[\psi(a_m)]^2 = \frac{p}{I(1-p)}\mathbb{E}[\psi(a_m)]^2,$$

which can also be made arbitrarily small by choosing $I \in \mathbb{N}$ sufficiently large. We now show that the first term in equation 11 can well approximate our target variance function $h$.

$$\mathbb{V}[(\epsilon_a - \epsilon_b)\psi(a_v)] = \mathbb{V}[\epsilon_a - \epsilon_b]\mathbb{V}[\psi(a_v)] + \mathbb{V}[\epsilon_a - \epsilon_b]\mathbb{E}[\psi(a_v)]^2 + \mathbb{V}[\psi(a_v)]\mathbb{E}[\epsilon_a - \epsilon_b]^2 \tag{12}$$

$$= 2p(1-p)\mathbb{V}[\psi(a_v)] + 2p(1-p)\mathbb{E}[\psi(a_v)]^2 \tag{13}$$

Define

$$h' = \sqrt{\frac{h}{2p(1-p)}} + \delta',$$

for some $\delta' > 0$. Again applying lemma 9 (which we can do independently of the choice of $a_m$ since neurons influencing $a_v$ and $a_m$ are disjoint), for any $\gamma'' > 0$ we can choose $\|\mathbb{E}[a_v] - h'\|_\infty < \gamma''$

and $\|\mathbb{V}[a_v]\|_\infty < \gamma''$. The first term in equation 13 can be made arbitrarily small by choosing $\gamma''$ small enough. We can again apply lemma 10 so that $\|\mathbb{E}[\psi(a_v)] - h'\|_\infty = \mathcal{O}(\Delta'\sqrt{\gamma''}/(\delta' - \gamma''))$. We then bound the difference between the second term in equation 13 and our target variance function:

$$\left\|2p(1-p)\mathbb{E}[\psi(a_v)]^2 - h\right\|_\infty \leq \left\|\sqrt{2p(1-p)}\mathbb{E}[\psi(a_v)] + \sqrt{h}\right\|_\infty \left\|\sqrt{2p(1-p)}\mathbb{E}[\psi(a_v)] - \sqrt{h}\right\|_\infty \tag{14}$$

$$\leq \left(\left\|2\sqrt{h}\right\|_\infty + \left\|\sqrt{2p(1-p)}\mathbb{E}[\psi(a_v)] - \sqrt{h}\right\|_\infty\right) \left\|\sqrt{2p(1-p)}\mathbb{E}[\psi(a_v)] - \sqrt{h}\right\|_\infty \tag{15}$$

where equation 14 follows from sub-multiplicativity of the infinity norm. Expanding the second term in equation 15,

$$\left\|\sqrt{2p(1-p)}\mathbb{E}[\psi(a_v)] - \sqrt{h}\right\|_\infty = \sqrt{2p(1-p)}\left\|\mathbb{E}[\psi(a_v)] - h' + \delta'\right\|_\infty$$
$$= \mathcal{O}(\delta' + \Delta'\sqrt{\gamma''}/(\delta' - \gamma''))$$

By first choosing $\delta'$ sufficiently small, and then choosing $\gamma''$ depending on $\delta'$, we can make this error term arbitrarily small. Since all the other contributions to $\mathbb{V}[f]$ were made arbitrarily small, this allows us to set $\|\mathbb{V}[f] - h\| < \gamma$, for any $\gamma > 0$, completing the proof. $\square$

In order to provide an analogous construction for MCDO BNNs with more than 2 hidden layers, we note that the above proof only requires a BNN output with a universal mean function and an arbitrarily small variance function in lemma 9. Instead of a 1HL network, we can apply theorem 3 to construct a 2 or more hidden layer network to provide these mean and variance functions. The rest of the proof then follows as in the 2HL case.

## E  Dropout With Inputs Dropped Out

The behaviour of MC dropout with inputs dropped out is somewhat different, both theoretically and empirically, from the case when inputs are not dropped out as discussed in the main body.

### E.1  Single-Hidden Layer Networks

In the single-hidden layer case, the variance is no longer convex as a function of $\mathbf{x}$. On the other hand, this approximating family still struggles to represent in-between uncertainty:

**Theorem 5** (MC dropout, dropped out inputs). *Consider a single-hidden layer ReLU neural network mapping from $\mathbb{R}^D \to \mathbb{R}^K$ with $I \in \mathbb{N}$ hidden units. The corresponding mapping is given by $f^{(k)}(\mathbf{x}) = \sum_{i=1}^{I} w_{k,i}\psi\left(\sum_{d=1}^{D} u_{i,d}x_d + v_i\right) + b_k$ for $1 \leq k \leq K$, where $\psi(a) = \max(0,a)$. Assume $\mathbf{v}$ is set deterministically and*

$$q(\mathbf{W}, \mathbf{b}, \mathbf{U}) = q(\mathbf{U})q(\mathbf{b}|\mathbf{U})\prod_i q_i(\mathbf{w}_i|\mathbf{U}),$$

*where $\mathbf{w}_i = \{w_{k,i}\}_{k=1}^{K}$ are the weights out of neuron $i$, $\mathbf{b} = \{b_k\}_{k=1}^{K}$ are the output biases and $q(\mathbf{U})$, $q(\mathbf{b}|\mathbf{U})$ and $q_i(\mathbf{w}_i|\mathbf{U})$ are arbitrary probability densities with finite first two moments. Then, for any finite set of points $\mathcal{S} \subset \mathbb{R}^D$ such that $\mathbf{0}$ is in the convex hull of $\mathcal{S}$,*

$$\mathbb{V}[f^{(k)}(\mathbf{0})] \leq \max_{\mathbf{s} \in \mathcal{S}}\left\{\mathbb{V}[f^{(k)}(\mathbf{s})]\right\} \quad \text{for} \quad 1 \leq k \leq K. \tag{16}$$

In order to prove theorem 5 we use the following simple lemma,

**Lemma 11.** *Let $f : \mathbb{R}^D \to \mathbb{R}$ be a convex function and consider a finite set of points $\mathcal{S} \subset \mathbb{R}^D$. Then for any point $\mathbf{r}$ in the convex hull of $\mathcal{S}$, $f(\mathbf{r}) \leq \max_{\mathbf{s} \in \mathcal{S}}\{f(\mathbf{s})\}$.*

*Proof of lemma 11.* Let $\{\mathbf{s}_n\}_{n=1}^{N} = \mathcal{S}_N \subset \mathbb{R}^D$. We proceed by induction. The lemma is true for $N = 2$ by the definition of convexity. Assume it is true for $N$. Let $\text{Conv}(\mathcal{S}_{N+1})$ denote the convex hull of $\mathcal{S}_{N+1}$. Consider a point $\mathbf{r}_{N+1} \in \text{Conv}(\mathcal{S}_{N+1})$. Then

$$f(\mathbf{r}_{N+1}) = f\left(\sum_{n=1}^{N+1} \alpha_n \mathbf{s}_n\right) \tag{17}$$

with $\sum_{n=1}^{N+1} \alpha_n = 1$ and $\alpha_n \geq 0$ for $1 \leq n \leq N+1$. We can write

$$f(\mathbf{r}_{N+1}) = f\left(\left(\sum_{n=1}^{N} \alpha_n\right)\mathbf{t}_N + \alpha_{N+1}\mathbf{s}_{N+1}\right) \leq \max\{f(\mathbf{t}_N), f(\mathbf{s}_{N+1})\} \qquad (18)$$

where $\mathbf{t}_N := \sum_{n=1}^{N} \alpha_n \mathbf{s}_n \big/ \sum_{n=1}^{N} \alpha_n$, and we have used the convexity of $f$. By the induction assumption, $f(\mathbf{t}_N) \leq \max_{\mathbf{s}\in\mathcal{S}_N}\{f(\mathbf{s})\}$, since $\mathbf{t}_N \in \mathrm{Conv}(\mathcal{S}_N)$. Combining this with equation 18 completes the proof. $\qquad\square$

*Proof of theorem 5.* By the law of total variance,

$$\mathbb{V}[f^{(k)}(\mathbf{x})] = \mathbb{E}[\mathbb{V}[f^{(k)}(\mathbf{x})|\mathbf{U}]] + \mathbb{V}[\mathbb{E}[f^{(k)}(\mathbf{x})|\mathbf{U}]].$$

Using lemma 1, $\mathbb{V}[f^{(k)}(\mathbf{x})|\mathbf{U}]$ is convex as a function of $\mathbf{x}$. As the expectation of a convex function is convex, the first term is a convex function of $\mathbf{x}$. This implies

$$\mathbb{E}[\mathbb{V}[f^{(k)}(\mathbf{0})|\mathbf{U}]] \leq \max_{\mathbf{s}\in\mathcal{S}}\left\{\mathbb{E}[\mathbb{V}[f^{(k)}(\mathbf{s})|\mathbf{U}]]\right\},$$

by lemma 11. $\mathbb{V}[\mathbb{E}[f^{(k)}(\mathbf{x})|\mathbf{U}]]$ is non-negative everywhere. As the output of the first layer is independent of the matrix $\mathbf{U}$ at $\mathbf{x} = \mathbf{0}$, $\mathbb{E}[f^{(k)}(\mathbf{0})|\mathbf{U}]$ is deterministic. So $\mathbb{V}[\mathbb{E}[f^{(k)}(\mathbf{0})|\mathbf{U}]] = 0$, completing the proof. $\qquad\square$

Figure 8: Schematic illustration of the bound in theorem 2, showing the input domain of a single-hidden layer MC dropout BNN, for the case $\mathbf{x} \in \mathbb{R}^2$. The convex hull (in blue) of the blue points contains the origin. Hence the variance at the origin cannot exceed the variance at any of the blue points.

## E.2 Deep Networks

In the case when the network has several hidden layers, dropout with inputs dropped defines a posterior with somewhat strange properties, as observed in Gal [4, Section 4.2.1]. In particular, in $D$ dimensions, a typical sample function from the approximate posterior will be constant as a function of roughly $pD$ of the input dimensions. However, which dimensions it is constant along depends on the particular sample. This behaviour is unlikely to be shared by the exact posterior. We are able to exploit this type of behaviour to show that if inputs are dropped out, there are simple combinations of mean and variance functions that cannot be simultaneously approximated by the corresponding approximating family.

**Proposition 1.** *Consider $f$ the (stochastic) output of an MC dropout network of arbitrary depth with inputs dropped out. For any $x, x' \in \mathbb{R}$ such that $\mathbb{V}[f(x)], \mathbb{V}[f(x')] < \epsilon^2$, $|\mathbb{E}[f(x)] - \mathbb{E}[f(x')]| \leq 2\epsilon\sqrt{2/p}$.*

*Proof.* With probability $p$, the input is dropped out, so $\Pr(f(x) = f(x')) \geq p$. We apply Chebyshev's inequality giving the bounds,

$$\Pr(|f(x) - \mathbb{E}[f(x)]| \leq r\epsilon) \geq 1 - 1/r^2 \quad \text{and} \quad \Pr(|f(x') - \mathbb{E}[f(x')]| \leq r\epsilon) \geq 1 - 1/r^2.$$

for any $r > 0$. Choose $r = \sqrt{2/p} + \delta$ for any $\delta > 0$, then there exists a realisation of the dropout network such that $|f(x) - \mathbb{E}[f(x)]| \leq r\epsilon$, $|f(x') - \mathbb{E}[f(x')]|$ and $f(x) = f(x')$ simultaneously.

Consequently,

$$
\begin{aligned}
|\mathbb{E}[f(x)] - \mathbb{E}[f(x')]| &= |\mathbb{E}[f(x)] - f(x) + f(x) - \mathbb{E}[f(x')]| \\
&= |\mathbb{E}[f(x)] - f(x) + f(x') - \mathbb{E}[f(x')]| \\
&\leq |\mathbb{E}[f(x)] - f(x)| + |f(x') - \mathbb{E}[f(x')]| \\
&\leq 2r\epsilon = 2\epsilon\sqrt{2/p} + 2\epsilon\delta.
\end{aligned}
$$

Taking the limit as $\delta \to 0$ completes the proof. $\qquad\qquad\square$

In other words we can bound the difference in the mean output at two points in terms of the uncertainty at those points and the dropout probability.

In $D > 1$ dimensions, we can get similarly tight bounds on lines parallel to a coordinate axis: for $\mathbf{x}, \mathbf{x}'$ on such a line $\Pr(f(\mathbf{x}) = f(\mathbf{x}')) \geq p$ still holds. If the dimension on which $\mathbf{x}$ and $\mathbf{x}'$ differ is dropped out $f(\mathbf{x}) = f(\mathbf{x}')$.

Alternatively in $D$ dimensions for arbitrary $\mathbf{x}, \mathbf{x}' \in \mathbb{R}^D$, $\Pr(f(\mathbf{x}) = f(\mathbf{x}')) \geq p^D$. This comes from noting that with probability $p^D$ the output of the network is a constant function. However, we note this bound becomes exponentially weak as the input dimension increases.

### E.3 Details of Experiments Minimising Squared Loss

We generated a dataset that consisted of two separated clusters in one dimension. We then fit a Gaussian process to the dataset and computed the predictive mean and variance on a one-dimensional grid of $N = 40$ points, call these point $X$. Let $\mu(X) \in \mathbb{R}^N$ denote the mean of the GP regression at these points $\sigma^2(X) \in \mathbb{R}^N$ denote its variance. We define a loss function as

$$
\mathcal{L}(\phi) = \|\mathbb{E}_{q_\phi}[f(X)] - \mu(X)\|_2^2 + \|\mathbb{V}_{q_\phi}[f(X)] - \sigma^2(X)\|_2^2.
$$

The expectation and variance are Monte Carlo estimated using 128 samples. We use full batch optimisation with ADAM with learning rate $1 \times 10^{-3}$ for 50,000 iterations. A dropout rate of $0.05$ is used for MCDO. Weights and biases are initialized at the prior for MFVI.

## F Details and Additional Figures for Section 4.2

In this appendix, we provide details of the protocol used to generate figure 5.

### F.1 Experimental Details

**Data:** We consider the dataset from figure 3 with $\mathbf{x} \in \overrightarrow{\mathbf{pq}}$, where $\mathbf{p} = (-1.2, -1.2)$ and $\mathbf{q} = (1.2, 1.2)$ i.e. the line segment between and including the two data clusters. We evaluate the overconfidence ratio on a discretisation of $\overrightarrow{\mathbf{pq}}$.

**Choosing the Prior:** For each depth a fully-connected ReLU network with 50 hidden units per layer is used. The prior mean for all parameters is chosen to be 0. The prior standard deviation for the bias parameters is chosen as $\sigma_b = 1$ for all experiments. In figure 5, the prior weight standard deviation is selected so that the prior standard deviation in function space at the region containing data is approximately constant. In particular, let $\sigma_w/\sqrt{H}$ be the prior standard deviation of each weight, where $H$ is the number of inputs to the weight matrix. We choose $\sigma_w = \{4, 3, 2.25, 2, 2, 1.9, 1.75, 1.75, 1.7, 1.65\}$ for depths 1-10 respectively, which ensures the prior standard deviations (of both the GP and the BNN) in function space at the points $(1, 1)$ and $(-1, -1)$ (the centres of the data clusters) are between 10 and 15. Choosing a fixed $\sigma_w$ such as $\sigma_w = 4$ for all depths would have caused the prior standard deviation in function space to grow unreasonably large with increasing depth; see Schoenholz et al. [10]. All models used a fixed Gaussian likelihood with standard deviation $0.1$.

**Fitting the GP:** The Gaussian process was implemented using GPFlow [9] with the infinite-width ReLU BNN kernel implemented following [7]. All hyperparameters were fixed and exact inference was performed using Cholesky decomposition.

Figure 9: Boxplots of the overconfidence ratios of HMC, MFVI and MCDO relative to exact inference in an infinite width BNN (GP) with $\sigma_w = \sqrt{2}$.

**Fitting MFVI:** We initialize the standard deviations of weights to be small and train for many epochs, following Tomczak et al. [12], Swaroop et al. [11] who found this led to good predictive performance. The weight means in each weight matrix were initialised by sampling from $\mathcal{N}(0, 1/\sqrt{2n_{\mathrm{out}}})$, where $n_{\mathrm{out}}$ is the number of outputs of the weight matrix. The weight standard deviations were all initialised to a very small value of $1 \times 10^{-5}$, (we tried a larger initialization with weight standard deviations initialized to $1 \times 10^{-2.5}$ and found no significant difference). Bias means were initialised to zero, with the variances initialised to the same small value as the weight variance. 100,000 iterations of full batch training on the dataset were performed using ADAM with a learning rate of $1 \times 10^{-3}$. The ELBO was estimated using 32 Monte Carlo samples during training. The local reparameterisation trick was used [6]. The predictive distribution at test time was estimated using 500 samples from the approximate posterior.

**Fitting MCDO:** The weights and biases were initialised using the default `torch.linear` initialisation. The dropout rate was fixed at $p = 0.05$. The $\ell^2$ regularisation parameter was set following Gal [4, Section 3.2.3] for the given prior, in such a way that the 'KL condition' is met, in the interpretation of dropout training as variational inference. 100,000 iterations of full batch training on the dataset were performed using ADAM with a learning rate of $1 \times 10^{-3}$. The dropout objective was estimated using 32 Monte Carlo samples during training. The predictive distribution at test time was estimated using 500 samples from the approximate posterior.

**Fitting HMC:** For HMC on the 1HL BNN, 250,000 samples of HMC were taken using the NUTS implementation in Pyro [5, 2] after 10,000 warmup steps. For the 2HL case, 1,000,000 samples of HMC were taken after 20,000 warmup steps. We set the maximum tree depth in NUTS to 5, and adapt the step size and mass matrix during warmup.

### F.2 Additional Figures

In order to assess the robustness of our findings to different choices of prior, we also consider the same experiment with $\sigma_w = \sqrt{2}$ for all depths. We choose this prior as it leads to similar variances in function space as depth increases [10]. We note that the variance of this prior is significantly smaller than the variance of the prior in the previous setting. The corresponding box plot is shown in figure 9. With this prior both methods tend to be less over-confident between data clusters, but more underconfident at the data, especially as depth increases (see figure 10).

## G  Initialisation of VI

In order to assess whether the variational objective (ELBO) or optimisation is primarily responsible for the lack of in-between uncertainty when performing MFVI and MCDO, we considered the effect of initialisation on the quality of the posterior obtained after variational inference. In order to find setting of the weights so that the posterior distribution in function space was close to the exact posterior in function space, we initialised the weights of the network by training the network using mean squared loss between the mean and variance functions of a reference posterior and the approx-

(a) Mean Field VI

(b) MC Dropout

(c) Mean Field VI ($\sigma_w = \sqrt{2}$ prior)

(d) MC Dropout ($\sigma_w = \sqrt{2}$ prior)

Figure 10: Plots of the overconfidence ratio against $\lambda$ (where $\lambda$ is defined as in figure 3) for several depths of neural networks with $\sigma_w = 4, 2, 1.7$ for 1, 5 and 9 hidden layers respectively (top), $\sigma_w = \sqrt{2}$ for all depths (bottom). Projections of the datapoints onto the diagonal slice between the clusters are shown as black crosses (✗). We see that both MCDO and MFVI are overconfident ($> 1$) in between data, and underconfident ($< 1$) at the locations where we have observed data, relative to the GP reference.

imate posterior (as in figure 4). The reference posterior was obtained by fitting the limiting GP on the dataset (shown in crosses). We used these weights as an initialisation for variational inference. The noise variance was fixed to the true noise variance that generated the data. The data itself was sampled from the limiting GP prior, so that the model should be able to fit the data well.

Two-hidden layer MFVI and MCDO networks were used, with 50 hidden units in both layers. The solution found by minimising squared loss for 50,000 iterations between the mean and variance functions and a reference posterior may lead to distributions over weights such that the KL from these distributions to the prior is high. This can lead to very high values of the variational objective function. To alleviate this behaviour, we gradually interpolate between the squared-error loss and the variational objective, by taking convex combinations of the losses. Call the function space squared loss $L_1$ and the standard variational objective $L_2$. Then after the first 50,000 iterations of using $L_1$, we train for 10,000 iterations using $.9L_1 + .1L_2$, 10,000 iterations using $.8L_1 + .2L_2$ and so on until we are only training using $L_2$. We then train for 100,000 iterations using just $L_2$, to ensure the variational objective has converged. Figure 11 shows that the obtained posterior still lacks in-between uncertainty, providing evidence that this may be due to the objective function rather than overfitting.

## H  Details and Additional Plots for Active Learning

### H.1  Experimental Setup

We use the same initialisation as in appendix F. As the dataset has low noise, we use a homoskedastic Gaussian noise model with a fixed standard deviation of $0.01$ for all models. We used the ADAM optimiser with learning rate $1 \times 10^{-3}$ for 20,000 epochs to optimise both MFVI and MCDO. We perform full batch training. All BNNs are retrained from scratch after the acquisition of each point from the pool set. We used 32 Monte Carlo samples from $q_\phi$ to estimate the objective function for

|(a) Mean Field VI|(b) MC Dropout|

Figure 11: Mean and error bars ($\pm$ 2 standard deviations) for the GP and the BNN with each inference scheme, trained on the data shown by the red crosses. The inference algorithms were initialised by first minimising the squared error to the reference GP mean and variance, and then running the respective inference algorithm.

both MFVI and MCDO. All networks had 50 neurons in each hidden layer. The prior for all BNNs and the GP was chosen to have $\sigma_w = \sqrt{2}, \sigma_b = 1$ (see appendix F for definitions). $\sigma_w = \sqrt{2}$ was chosen so that the prior in function space has a stable variance as depth increases [10]. The dropout probability was set at $p = 0.05$ for all MCDO networks. The dropout $\ell_2$ regularisation was chosen to match the 'KL condition' as stated in Gal [4, Section 3.2.3]. The results are averaged over 20 random initialisations and selections of the 5 initial points in the active set. For MFVI and MCDO, the predictive distribution at test time and the predictive variances used for active learning were estimated using 500 samples from the approximate posterior. The parameter initialisations are the same as those in appendix F.

## H.2 Additional Figures

Figure 12 shows the points chosen by deep BNNs. Again the GP chooses points from every cluster, and seems to focus on the 'corners' of each cluster. MFVI samples from more clusters than the 1HL case, but still comparatively oversamples clusters further from the origin, and undersamples those near the origin. MCDO has a more spread out choice of points than the 1HL case, but still fails to obtain significantly better RMSE than random.

Figures 13 and 14 show the predictive uncertainty of 1HL models at the beginning and end of active learning. All models significantly reduce their uncertainty around clusters that have been heavily sampled, except for MCDO. This causes MCDO to repeatedly sample near locations that have already been labelled, in contrast to the GP. Note also that MFVI is most confident at clusters near the origin that have never been sampled, and least confident at clusters far from the origin that have already been heavily sampled.

(a) Limiting GP        (b) MFVI        (c) MCDO

Figure 12: Points chosen during active learning in the 3HL case. Colours denote distance from the origin in 14-dimensional input space. Grey crosses (✕) denote the five points randomly chosen as an initial training set. Red crosses (✕) denote the 50 points selected by active learning. Again, the GP samples the corners of each cluster, and MFVI oversamples clusters far from the origin.

(a) Limiting GP        (b) MFVI        (c) MCDO

Figure 13: Colours denote *predictive uncertainties* in the 1HL case, at the beginning of active learning. As the noise standard deviation was fixed to 0.01 for all models, changes in the predictive standard deviation reflect model uncertainty. Grey crosses (✕) denote the five points randomly chosen as an initial training set.

(a) Limiting GP        (b) MFVI        (c) MCDO

Figure 14: Colours denote *predictive uncertainties* in the 1HL case, after 50 iterations of active learning. As the noise standard deviation was fixed to 0.01 for all models, changes in the predictive standard deviation reflect model uncertainty. Grey crosses (✕) denote the five points randomly chosen as an initial training set. Red crosses (✕) denote the 50 points selected by active learning. Note how, compared to figure 13, the GP has reduced its uncertainty near points it has observed, and is most uncertain on corners opposite those points. In contrast, for both MFVI and MCDO, the network is still uncertain around regions it has already collected points from, leading it to oversample those clusters and undersample others.