[Reviews · NeurIPS 2020]

Review 1

Summary and Contributions: This paper provides a new perspective on typical inference methods (mean-field VI and Monte Carlo drop out) for BNNs. It shows that those methods have a fundamental limitation in representation uncertainty between data points.

Strengths: * The paper discusses an important shortcoming of typical inference methods for BNNs and may inspire research that overcomes these limitations. * The contribution is novel and is of interest to the NeurIPS community.

Weaknesses: * The work seems limited to the regression case. Could you discuss an extension to classification?

Correctness: * All claims and derivations seem to be correct. * The empirical evaluation seems sound but could be a bit more thorough.

Clarity: The paper is well written and easy to follow.

Relation to Prior Work: The related work is clearly discussed.

Reproducibility: Yes

Additional Feedback: * Do you believe that your target posterior (using a Gaussian prior) actually represents uncertainty well? Recent work showed limitations of the Bayesian posterior in BNNs using Gaussian priors. * It would be great if you could discuss what properties the variational family needs to meet to represent in-between uncertainty correctly. UPDATE AFTER REBUTTAL The authors addressed my questions and I have decided to keep my score of 6.


Review 2

Summary and Contributions: The authors consider two claims about approximate posterior distributions: 1-The approximating family contains good approximations to the true posterior. 2-The approximate inference method used must be able to find them. They prove that: -For single-hidden-layer models there are simple settings where no fully-factorized Gaussian or MC dropout distribution can be a good fit. -For deeper models, Criterion 1 is satisfied, but the authors offer evidence that deep BNNs still have similar pathologies affecting Criterion 2.

Strengths: -Theorems 1 and 2 are well presented and interesting. The authors acknowledge that the specific situations that they cover might be limited, but provide some evidence in Appendix A that MFVI/MCDO might be bad in more general cases. -Theorem 3 is similarly good. -A number of the experiments are clever and well executed and serve to distinguish between different related hypotheses.

Weaknesses: -I would be inclined to move Figure 9 to the main body. It's actually very interesting that a deeper network is able to pass this test. It puts the emphasis of your argument in section 3.2 onto the claim that it is the inference method, not the model, which is problematic. -Relatedly, I think this interpretation could be clearer in the introduction. In the paragraph starting line 49 you could be clearer that Theorem 3 is a result about models, not inference, and that you have other evidence that the deeper models are fine, but that there are problems with inference. -Your title also seems to be affected by a confusion here. Part of your work is about the expressiveness of approximate *posteriors* and part of it is about the success of approximate *inference*. -I think you slightly overstate the "emprirical evidence that in spite of this flexibility VI in deep BNNs still leads to distributions that suffer from similar pathologies to the shallow case". Your experiments are on fairly small data compared to typical deep learning practices, and you should probably acknowledge that a number of authors have gotten pretty good results with both of these inference methods. I think you can make this statement slightly weaker without detracting from the importance of your work but bringing it more in line with what your experiments actually show. Minor: -Note that Figures 6 and 7 are useful for existence claims but are being used here for for all claims. That is, if you can show an uncertainty plot that is good, you show that a method can be good. If you show one that is bad, you do not show that they cannot be good. This makes Figure 7 less clear cut, especially if you do not describe the range of hyperparameters considered.

Correctness: The results in Table 1 look off to me. Wile it is possible that the deeper models with random acquisition just perform worse it seems more likely that the hyperparameters need to be separately tuned for shallower and deeper models, and it seems you are still using the same hyperparameters for all models. I think this paper is impressive enough to be accepted anyhow, but if you were able to improve on this experiment I think it might help you feel more confident that your worries about inference in deeper models apply in practice.

Clarity: Yes. The paper is a bit dense at points, but it is well written and does a good job with proof sketches to make otherwise complicated proofs interpretable.

Relation to Prior Work: It would be sensible to acknowledge that a number of authors have had considerable success with deep VI/MCDO models, and that in relation to those your results in section 5 should really be seen as raising a question about the inference rather than offering an answer to Criterion 2.

Reproducibility: Yes

Additional Feedback: Section 2.2 does not seem like an important point for you to make. There is extensive prior work (as you acknowledge) on BNN priors and this is not core to your argument. It's probably enough to acknowledge somewhere in a sentence that you pick priors that are not perverse and create space for figures that are more important to your paper that have been moved to the appendix. ---------------------- Thanks for your author response, and for engaging so swiftly with my question about the baseline. I really like this paper. That said, I do think you imply that your results for deeper networks are stronger than they actually are. I think there's a good chance that the effects you identify are much more pronounced for small numbers of datapoints in low-dimensional data. This is reinforced by the observation that your Naval results are quite sensitive to using 55 datapoints instead of the full set. The deeper model performs worse for small amounts of data, but not for larger, suggesting it is overparameterized/underregularized. This makes it unclear that the effects at depth that you identify have the same source as the effects in shallow models that you identify. Naval is also a slightly odd dataset, which single-layer NNs can almost exactly fit, whereas other datasets do not have this problem, did you try other UCI datasets? Also, I think sample-based MI estimators for regression can be quite bad. Basically, I remain quite sceptical about this experiment. I think you should either, then, do experiments in a setting with lots of high-dim data or which otherwise resolves these issues (not at all necessary to achieve publication) or make it clearer that there are limitations to the evidence that your single-layer results extend to deeper models. I've heard some people who have read the paper summarizing it as having said things like "MFVI/MCDO has pathology X" and not differentiating the single-layer and deeper settings, which I think is partly helped by some of the current framing. I'm giving an accept score on the strength of the theoretical results, and the creativity of the experiments, but I'd be disappointed if the mismatch between implied conclusions and the experiments provided were not addressed for the camera ready version.


Review 3

Summary and Contributions: The paper analyzes the issues of the mean field variational inference and Monte-Carlo dropout in uncertainty quantification for Bayesian neural networks. The paper points out that for one layer NN, MFVI and MCDO cannot increase the uncertainty for the prediction in between two separate regions of low uncertainty, and for deep NNs, the issues can be overcome in theory. But the paper empirically shows similar issues (less severe but still there) in learning deep NNs.

Strengths: 1. a very important topic. Uncertainty quantification should be a central task in Bayesian learning, but is largely ignore by the community. 2. profound analysis. and solid results. 3. interesting conclusions

Weaknesses: It will be good to discuss some ideas about how to address or alleviate the issue of MFVI and MCDO in uncertainty estimation for BNNs.

Correctness: yes

Clarity: yes

Relation to Prior Work: yes

Reproducibility: Yes

Additional Feedback:


Review 4

Summary and Contributions: My takeaway from the paper: The paper reports that factorizing Gaussian posterior, a common choice in BNNs, has a problem of underestimating the variance around the area where there are not many observations. Under a single hidden layer neural network model, this behaviour is severe as shown in Figure 2 and 3. With more layers of a model, the behaviour becomes less severe although it persists.

Strengths: The strengths I believe the paper has: 1. While it is well known that factorizing Gaussian posterior typically underestimates the posterior variance (i.e., overconfident), this paper seems to formally state it in Theorem 1. 2. Observing this behaviour in deep networks seems to be adding a stepping stone in the study of BNNs, as illustrated in Figure 4 & 5 and Table 1.

Weaknesses: The paper's weaknesses I found: 1. While it's nice to see a formal statement of the behaviour of underestimating variance (i.e., over-confident) under the factorizing Gaussian posterior combined with variational inference, this is something known. In fact, many entry level machine learning text books contain this, e.g., take a look at C.Bishop about approximate inference. Also, Renyi divergence VI paper by Li and Turner also states the pathology of the mean field Gaussian posterior under VI (alpha goest to 1 in Renyi divergence) being overconfident in Figure 1. I understand that these results are shown under the shallow models, e.g., Bayesian linear regression, logistic regression, etc. The proposed result in the single-layer neural net could be viewed as a type of shallow model with a specific link function (nonlinearity). Hence, I don't really see such a novelty in their findings reported in the submitted paper. 2. Another weakness I found was it is not so clear to me how these squashed variance between the space where there are not many observations changes with a different objective function (not just a squared difference) and different nonlinearities in the shallow and deep networks, e.g., the shallow network examples shown in Figure 2 and 3. Is the flatness of the variance in the in-between regions a result of ReLu? Also, is the symmetry in the resulting variance estimate a result of squared error in the function? If one chooses to use different nonlinearities and losses, would this behaviour also remain the same? If not, what would change? Are there any setups where this pathological behaviour is less severe? I have read the rebuttal, and would like to keep my score the same, as the novelty aspect seems weak.

Correctness: The claim seems to be correct, while I didn't go through all the steps in their proofs.

Clarity: Yes, I think this paper is clearly written.

Relation to Prior Work: Yes, the relation to the previous work was clearly and correctly mentioned.

Reproducibility: Yes

Additional Feedback:

[Author Response · NeurIPS 2020]

We thank the reviewers for their feedback. All reviewers found the exposition clear and most agreed the results were novel and of interest to the Bayesian machine learning community. As suggested, we will add an expanded broader impact statement, examining the role of uncertainty quantification in high-risk and scientific applications.

**R1. Potential extension to classification:** Thms 1-3 apply to the variance of the logits in classification. However, an extension of Thms 1-2 to classifier uncertainty is not straightforward since that also depends on the logit means.

**R1. Justification of the chosen prior:** HMC and the limiting GP provide good in-between uncertainty (Fig 3) and active learning performance (Sec 5, Table 1). This means the chosen prior encodes reasonable assumptions for the low dimensional regression tasks we consider, and the poor performance of MFVI/MCDO is due to bad inference. In contrast, some recent work examining BNN priors, e.g. Wenzel et al. 2020, considers image classification with Bayesian ResNets, which is a different probabilistic model and a different task.

**R2. Clarification on distinction between model and inference:** We will move Fig 9 into the text to illustrate Thm 3, and emphasise in the introduction that Criterion 1 *is* satisfied for deep BNNs for MFVI and MCDO posteriors (lines 168-170, 264). We may be using the term "model" differently from R2: we mean "probabilistic model" i.e. a prior and likelihood. We therefore see Thm 3 as a result about inference rather than a model (since it concerns the form of an approximate posterior, rather than the exact posterior) — we will clarify this.

**R2. Clarifying limitations of empirical evidence:** We will state on lines 51-53 that our experiments focus on the small-data regime and low-dimensional regression, where comparison with exact inference is easier to perform. Although previous authors have obtained good empirical results on downstream tasks (as mentioned on line 23), previous work does not generally focus on how well the approximate predictive resembles the exact one, as we do (lines 296-298).

**R2. Potential sub-optimality of hyperparameters in active learning experiments (Table 1):** Following the review, we performed some manual hyperparameter tuning for the prior & dropout rate for MCDO Random (validating on the test set). This brought 4HL RMSE from $0.443 \pm 0.01$ to $0.387 \pm 0.02$, but this result is still worse than the 1HL case. More extensive search may be able to improve this further, but extensive hyperparameter search is generally impractical in online active learning. Our main goal in Sec 5 was to evaluate the quality of approximate inference compared to the limiting GP (which is closer to exact inference), rather than to improve active learning performance in general. The GP performs significantly better than random selection for all depths, meaning that in the small data regime, the benefit of this Bayesian prior combined with accurate inference is clear. In comparison, the poor results of MFVI & MCDO suggest the worse performance is mainly due to the bad approximate inference in deep BNNs (lines 257-259).

**R3. Potential directions on improving uncertainty quantification (UQ):** For 1HL BNNs new objective functions with mean-field Gaussian families will not solve issues regarding UQ (Thms 1,2). For deep BNNs Thm 3 tells us that MFVI/MCDO are sufficiently expressive for many tasks that rely on UQ, so improved objective functions may lead to improvements (e.g. Sun et al. 2019 [37], Fig 1).

**R4."Lack of novelty":** We respectfully disagree. To the best of our knowledge, Thms 1-3 are the first theoretical results on the quality of BNN approximate inference in terms of *estimating function space uncertainty*. The derivations are non-trivial, and the results apply *regardless* of the inference algorithm (not just VI, see lines 73-75, 118 & 130-131). This includes methods which are usually not expected to be over-confident, e.g. EP and Rényi VI, as long as factorised Gaussians/dropout distributions are used as approximate posteriors.

**R4. Parameter vs function space:** The over-confidence of VI in *parameter space* is well-known. However, it is not obvious how this translates into *function space*. Our results focus on uncertainty in function space (lines 35-36, 144-145). To illustrate this difference we run mean-field VI for a simple Bayesian linear regression (BLR) model with RBF features, defined by $y(x) = \sum_{i=-10}^{10} w_i \psi_i(x)$, $\psi_i(x) = \exp(-(x-i)^2)$, with $\mathcal{N}(0,1)$ priors on $w_i$. Even though MFVI is over-confident in parameter space (Fig 1 bottom), it still shows significant in-between uncertainty in function space (Fig 1 top). This shows that weight-space intuition does not necessarily translate to function space. The fact that 1HL MFVI & MCDO BNNs, usually thought of as more flexible than BLR, *cannot* represent this kind of uncertainty is non-obvious, and this is pointed out by our contributions.

Figure 1: BLR with RBF features & MFVI.

**R4. Why we minimise squared error in Fig. 2:** Thms 1 & 2 show that no factorised Gaussian or dropout posterior can give in-between uncertainty for 1HL ReLU BNNs, regardless of objective. To verify this, we find the closest approximation within the approximating family to a desired target (with in-between uncertainty) by directly minimising the squared error between the variance functions. The symmetry arises because we chose a symmetric target, not because of the squared error loss. We also optimise the ELBO (Fig 3c,d, Fig 4).

**R4. Extension to other non-linearities:** Thms 1 & 2 apply to ReLU non-linearities. Empirically, we observed that Tanh BNNs also struggle with in-between uncertainty, but we currently do not have a theoretical proof of this. Thm 3 can likely be extended to other non-linearities and we leave this to future work.

[Meta-Review · NeurIPS 2020]

After much discussion and follow up questions to the authors, the reviewers converged towards recommending to accept this submission. The reviewers were satisfied with the authors' response, and have updated their reviews accordingly. There are some remaining points about claims made in the paper which need to be toned down (single layer vs deep models), and the conclusions from empirical validation only supporting claims with small low dim data (while the effect reverses with the full 11000 datapoints in AL). I recommend acceptance and trust that the authors will address these remaining points for the camera ready.